# Numerical modelling of convective heat transport by air flow in permafrost talus slopes

Jonas Wicky[1], Christian Hauck[1]

[1]Department of Geosciences, University of Fribourg, Fribourg, 1700, Switzerland

*Correspondence to*: Jonas Wicky (jonas.wicky@unifr.ch)

**Abstract.** Talus slopes are a widespread geomorphic feature in the Alps. Due to their high porosity a gravity-driven internal air circulation can be established which is forced by the gradient between external (air) and internal (talus) temperature. The thermal regime is different from the surrounding environment, leading to the occurrence of permafrost below the typical permafrost zone. This phenomenon has mainly been analysed by field studies and only few explicit numerical modelling

studies exist. Numerical simulations of permafrost sometimes use parameterizations for the effects of convection, but mostly neglect the influence of convective heat transfer in air on the thermal regime. On the contrary, in civil engineering many studies have been carried out to investigate the thermal behaviour of blocky layers and to improve their passive cooling effect. The present study further develops and applies these concepts to model heat transfer in air flows in a natural scale talus slope. Modelling results show that convective heat transfer has the potential to develop a significant temperature

difference between the lower and the upper parts of the talus slope. A seasonally alternating chimney-effect type of circulation develops. Modelling results also show that this convective heat transfer leads to the formation of a cold reservoir in the lower part of the talus slope, which can be crucial for maintaining the frozen ground conditions despite increasing air temperatures caused by climate change.

## 1 Introduction

Mountain permafrost is currently undergoing substantial changes due to climate change as a whole and especially due to the observed air temperature increase. Among the typical mountain permafrost substrates, i.e. rock, fine sediments and coarse blocky surfaces, the latter have an important role on the subsurface because of their high insulating characteristics due to the low thermal conductivity of the air voids between the blocks (Harris and Pedersen 1998, Herz et al. 2003, Juliussen and Humlum 2008). In addition, natural air convection with upward transport of warmer air from the permafrost body and

downward transport of cold air from the surface can take place within the coarse blocky layer, both vertically (in flat terrain) as well as in form of a 2-dimensional circulation within a slope. These two effects firstly lead to much lower surface and subsurface temperatures for terrain with coarse blocky surface layers compared to fine-grained or bedrock surfaces (e.g. Hanson and Hölzle 2004, Schneider et al. 2012, Gubler et al. 2011), but then also to persisting permafrost occurrences at the lower limit of permafrost (including rock glaciers and ice-cored moraines) and even to azonal permafrost occurrences at low

elevation (e.g., Kneisel et al. 2000, Gude et al. 2003, Delaloye et al. 2003, Kneisel et al. 2015). For the latter, mean annual air temperatures may be positive and permafrost would not exist without this effect. This permafrost mostly occurs in undercooled scree/talus slopes and has been well researched in a number of publications (Funk and Hoelzle 1992, Wakonigg 1996, Kneisel et al. 2000, Gude et al. 2003, Sawada et al. 2003, Delaloye et al. 2003, Gorbunov et al. 2004, Delaloye and Lambiel 2005, Zacharda et al. 2007, Morard and Delaloye 2008, Morard et al. 2008, Phillips et al. 2009, Gądek and Leszkiewicz 2012, Stiegler et al. 2014, Kneisel et al. 2015).

Previous review articles about mountain permafrost research have highlighted the need for improved process understanding and model studies for the complex subsurface material of Alpine terrain (Harris et al. 2009, Haeberli et al. 2010, Etzelmüller 2013). While the energy balance for the coarse blocky surface layer of rock (and debris-covered) glaciers and talus slopes has been addressed in several field and modelling studies (e.g., Hanson and Hoelzle 2004, Scherler et al. 2014, Reid and Brock 2010), the effect of the internal 2-dimensional air circulation driven by the gradient between outside (air) and internal (talus) temperature has not been treated in detail or quantified with respect to the other terms in the energy balance. Transient numerical simulations of permafrost usually neglect the influence of convective heat transfer in air on the thermal regime or parameterize it with an apparent thermal conductivity (Gruber and Hoelzle 2008), or with an artificial heat sink (Scherler et al. 2013, 2014). Only very few and idealised modelling studies of the air circulation in undercooled talus slopes exist to date (e.g., Tanaka et al. 2000, 2006).

In contrast, many studies have been carried out in civil engineering to investigate the thermal behaviour of artificial blocky layers (such as railroad embankments) and to improve their passive cooling capacity (e.g., Goering and Kumar 1996, Cheng 2005, Cheng et al. 2007, Pham et al. 2008, Zhang et al. 2005, Arenson et al. 2006, 2007). In this study we will adapt the modelling concepts from civil engineering to investigate the feasibility of using a commercially available finite-element modelling software (GeoStudio 2013a, b, c) to explicitly model the 2-dimensional heat transfer by air flow within an idealised talus slope and to quantify the cooling effect with respect to different sensitivity parameters. The simulations are driven by observational data from the Lapires talus slope in the Western Swiss Alps (Delaloye and Lambiel 2005, Scapozza et al. 2015, Staub et al. 2015).

## 2 Theory and data

### 2.1 Conceptual model

Convection is the transport of energy by a fluid, in the present case the transport of heat by air flow through the available pore space. This is in contrast to heat transport by conduction or radiation, which are other important processes for the energy balance within coarse blocky surface layers (Scherler et al. 2014). Convective processes are responsible for the specific (cold) temperature regime in undercooled talus and scree slopes. In 1900 already, Balch described the asymmetry of seasonal air circulation in ice caves with gravity-induced descending cold air in winter, and an absence of circulating air in summer (Balch 1900). Wakonigg (1996) described the so-called chimney effect in talus slopes, where a seasonally switching

air circulation produces descending cold air in summer and ascending warm air in winter. The cold air exits at the foot of the slope in summer and warm air exits through melt holes in the snow cover at the top of the slope in winter, as has been observed in field studies on various talus and scree slopes (e.g., Gude et al. 2003, Sawada et al. 2003, Delaloye 2003, Delaloye et al. 2003, Zacharda et al. 2005, Zacharda et al. 2007, Morard et al. 2008, Lambiel and Pieracci 2008, Morard, 2011).

Figure 1 shows this process schematically in its most basic form (Morard et al. 2010). Here, the sediments in the talus slope are assumed to be homogeneous and the induced air circulation is most effective when the temperature gradient between the outside air ($T_{ao}$) and the voids in the porous substrate ($T_{ai}$) is high and if the voids between the blocks have a high connectivity. Material composition can be heterogeneous which means that the cooling effect does not affect the whole talus slope in the same way (Delaloye and Lambiel 2005). In any case, the internal air circulation can lead to rapid ground cooling, in particular at the bottom of the talus slope, where cold air is aspirated in winter.

Model studies trying to explicitly simulate this process are rare. In a 1-dimensional study of the energy balance within the coarse blocky surface layer of a rock glacier, Scherler et al. (2014) estimated the impact of this seasonally varying convective heat transfer (heat source in winter, heat sink in summer) in comparison with other elements of the energy balance, and therefore also with observed data. However, this approach does not treat the underlying process and prohibits an explicit modelling of the 2-dimensional air circulation. Tanaka et al. (2000) investigated the air circulation at two sites (Ice Valley, Korea, and Nakayama, Japan) using field data, and analytical and numerical models. In their most complex 2-dimensional model they could confirm that the cooling effect in summer stems from a gentle katabatic flow in stable stratified conditions, whereas the winter circulation is due to an unstable convective overturning in a 50 m long convection cell. In general, these convection cells form by natural convection, i.e. air movement occurs as a function of density (temperature) differences, as opposed to forced convection caused by the influence of surface (atmospheric) wind (e.g., Arenson and Sego 2007). Hereby, the air movement due to natural convection has to be strong enough with respect to the bulk thermal conductivity of the material to yield a sustained convective cell. This relation is expressed by the Rayleigh number, which relates the air permeability, the thickness of the porous layer, the thermal conductivity of the material and the temperature gradient within the porous layer, i.e. the talus material. The higher the air permeability, the thickness of the talus and the temperature gradient with respect to the thermal conductivity, the higher the Rayleigh number and the stronger and longer the convective cell(s) (Arenson and Sego 2007). However, as some of the important parameters of the Rayleigh number may be temporally and spatially variable, the air circulation will change in both space and time. Consequently, an explicit modelling of the air circulation in 2 dimensions is necessary to be able to simulate the development and seasonality of the occurrence of convection cells in talus slopes. Convective heat transfer in permafrost was explicitly modelled in several engineering studies. First numerical simulations were conducted by Goering and Kumar (1996) and showed a passive cooling effect of convective cells within coarse grained road embankments. The same effect, corresponding to a Rayleigh-Bénard circulation was described by Guodong et al. (2007) as a thermal semi-conductor which cools road or train embankments in permafrost regions in winter, whereas no heat can penetrate in summer, due to the stable temperature

stratification within the coarse material. Further engineering studies (e.g., Arenson and Sego, 2006, Arenson et al. 2007, Pham et al. 2008, Pei et al. 2014) investigated and simulated these small-scale cooling effects to optimise passive cooling systems on mine waste piles, road- and railway embankments in permafrost regions.

In the following we will describe the model setup for an explicit simulation of the larger and natural scale lateral heat transfer by two-dimensional air circulation within an Alpine talus slope.

## 2.2 Governing equations

The general heat flow equation including convective transfer by water and air can be formulated as follows (GeoStudio 2013a):

$$\left(\rho_s c_{ps} + Lw\frac{\partial W_u}{\partial T}\right)\frac{\partial T}{\partial t} = \frac{\partial}{\partial y}\left[K_t \frac{\partial T}{\partial y}\right] + c_{pa}\frac{\partial(\dot{m}_a T)}{\partial y} + \rho_w c_{pw}\frac{\partial(q_w T)}{\partial y} + Q \tag{1}$$

where

| | |
|---|---|
| $L$ | latent heat of water [J m$^{-3}$] |
| $W_u$ | unfrozen water content [m$^3$ m$^{-3}$] |
| $w$ | volumetric water content [m$^3$ m$^{-3}$] |
| $T$ | temperature [K] |
| $y$ | coordinate(s) [m] |
| $K_t$ | thermal conductivity [W m$^{-1}$ K$^{-1}$] |
| $Q$ | boundary flux [W m$^{-3}$] |
| $t$ | time [s] |
| $\rho_{s/w}$ | density soil/ water[g m$^{-3}$] |
| $c_{ps}$ | specific heat capacity soil [J g$^{-1}$ K$^{-1}$] |
| $c_{pa/pw}$ | specific heat capacity air/water [J g$^{-1}$ K$^{-1}$] |
| $\dot{m}_a$ | mass flow air [g s$^{-1}$] |
| $q_w$ | flow rate (Darcy) of water [m s$^{-1}$] |

Water (Eq. 2) and air (Eq. 3) flow can be described as follows and have to be jointly solved, as air and water pressure are dependent variables (GeoStudio 2013b,c):

$$m_w \gamma_w \frac{\partial H_w}{\partial t} = \frac{\partial}{\partial y}\left[K_w \frac{\partial H_w}{\partial y}\right] + m_w \frac{\partial P_a}{\partial t} + Q_w \tag{2}$$

$$\left(\frac{\theta_a}{RT} + \rho_a m_w\right)\frac{\partial P_a}{\partial t} = \frac{\partial}{\partial y}\left[\frac{\rho_a K_a}{\gamma_{oa}}\frac{\partial P_a}{\partial y} + \frac{\rho_a^2 K_a}{\rho_{oa}}\right] - \left[\frac{\theta_a P_a}{R}\frac{\partial(\frac{1}{T})}{\partial t}\right] + \left[\rho_a \gamma_w m_w \frac{\partial H_w}{\partial t}\right] \tag{3}$$

where

$m_w$      *slope of the water storage curve* $[m\,s^2\,g^{-1}]$

$\gamma_w$      *unit weight of water* $[g\,m^{-2}\,s^{-2}]$

$\gamma_{oa}$      *relative unit weight of water* $[g\,m^{-2}\,s^{-2}]$

$H_w$      *total head* $[m]$

$K_w$      *hydraulic conductivity* $[m\,s^{-1}]$

$Q_w$      *boundary flux* $[s^{-1}]$

$K_a$      *air conductivity* $[m\,s^{-1}]$

$P_a$      *air pressure* $[Pa]$

$\rho_a$      *density of air* $[g\,m^{-3}]$

$\rho_{oa}$      *relative density of air* $[g\,m^{-3}]$

$\theta_a$      *volumetric air content* $[m^3\,m^{-3}]$

$R$      *specific heat capacity of dry air* $[J\,g^{-1}\,K^{-1}]$

## 2.3 Forcing data sets

Forcing data are used from the Lapires talus slope which is situated between 2400 and 2700 m a.s.l. in the Valais Alps, Western Swiss Alps (e.g., Staub et al. 2015). Several long-term mean annual ground near-surface temperature data series and four deep boreholes with temperature measurements are available from this site (Delaloye and Lambiel 2005, Scapozza 2013, Scapozza et al. 2015, Staub et al. 2015), which is part of the PERMOS network (PERMOS, 2016b).

For this study we used ground surface temperature (GST) data as the surface atmosphere boundary condition. Moreover, snow height data from the nearby weather station Les Attelas (Swiss IMIS network) was used to model the seasonal decoupling of the ground from the atmosphere by the snow cover. Borehole temperature data were used to compare the modelled temperature distribution to the measured temperature series in the borehole.

Validation and ground truth data are available from four boreholes, regarding temperature, material composition and stratigraphy of the sediments (Delaloye et al. 2001, Scapozza 2013, Scapozza et al. 2015), and from geophysical measurements for the analysis of spatial heterogeneity (Delaloye 2004, Lambiel 2006, Hilbich et al. 2008, Hilbich 2010). The porosity within the permafrost layer ranges from 30 to 60% and some parts of the talus slope (including the base of the slope) are sealed by ice (cf. Scapozza 2013, Hilbich 2010). The bedrock consists of gneiss and is located at 40 m depth at least, as observed in borehole cores (Scapozza 2013, Scapozza et al. 2015).

## 3 Model setup

The commercially available software GeoStudio has already been applied to coarse blocky permafrost substrates for of different engineering purposes, such as the investigation of the passive cooling effect in road embankments (Arenson et al. 2006) or the heat convection in coarse mine waste rock piles (Arenson et al. 2007). Furthermore, Mottaghy and Rath (2006) used it to validate their own model approach in a study of paleoclimate permafrost in porous media. In the engineering sciences modelling approaches of convective heat transport by air and the resulting passive cooling effect are numerous (e.g., Goering and Kumar 1996, Guodong et al. 2007, Lebeau and Konrad 2009). Such studies showed the important influence of convective heat transport on the ground thermal regime at fine scales.

In this study we pick up these concepts and adapt them to a natural scale talus slope. With GeoStudio we use a finite element modelling approach that solves the partial differential heat equations, including convective heat transfer over a 2-dimensional domain. The numerical domain is divided into sub-domains with different material properties (Fig. 2). Measured daily temperature values (see Sect. 2.3) were used as temperature forcing data at the surface-atmosphere boundary for the transient modelling. This 2-dimensional approach allows an explicit modelling of the air flow and the resulting convective heat exchange over the spatial domain of a natural scale talus slope, allowing to analyse its influence on the ground thermal regime.

### 3.1 Mesh geometry

Figure 2 shows the numerical domain with its four sub- domains. Each sub- domain represents a different material and therefore has different physical material properties (Sect. 3.2). The entire domain represents an idealised talus slope with a talus thickness of 17 m and a constant slope angle of 33°. An unstructured mesh of quadrilateral and triangular elements was used with a mesh size of 1.2 m in the talus increasing to 3 m in bedrock and air domain. An adaptive time stepping scheme was used. The maximum time step is 0.5 days, decreasing to 0.1 days until the Courant criterion is met to minimize numerical dispersion and oscillation (GeoStudio 2013a, c). Convergence is met when the results of two different iterations differ by less than 0.01°C or do not differ by more than 0.1% for temperature and 0.001% for pressure. The maximum number of iterations was set to 30, and our experiments showed that the model is numerically stable and produced interpretable results under these numerical conditions. Numerical configuration proved, however, to be a crucial and delicate point in defining the model boundaries and running the model.

### 3.2 Material properties

Table 1 summarises the material properties of the sub-domains shown in Fig. 2 which were used in our simulations. The conductivities and heat capacities are taken from Schneider (2014), who empirically established the different values for several blocky periglacial landforms in the Murtèl-Corvatsch region. It can be assumed that these values are more accurate for natural-scale talus slopes than published values from laboratory experiments. The volumetric heat capacity for bedrock

exposed at the surface is given by Schneider (2014) as 275 kJ m$^{-3}$K$^{-1}$ which is approximately eight times lower than other published values for deep seated bedrock in permafrost (e.g., 1600 kJ m$^{-3}$K$^{-1}$ in Gruber and Hölzle 2008, > 2000 kJ m$^{-3}$K$^{-1}$ in Arenson and Sego 2006, 2000 kJ m$^{-3}$K$^{-1}$ in Nötzli et al. 2008, 2063 kJ m$^{-3}$K$^{-1}$ in Wegmann et al. 1998). Considering that the model is simulating bedrock at depth and not at the surface, the volumetric heat capacity is set to 2500 kJ m$^{-3}$ K$^{-1}$. Arenson et

al. (2006) point out that the heat capacity has a less prominent influence on the temperature distribution than other parameters in Table 1, especially the air conductivity.

The porosity is set to 0.5 for the material within the talus slope. This value corresponds to measured porosity values in the drill cores (Scapozza et al. 2015) and model fitted values for the Lapires site (Marmy et al. 2016) as well as previously published values for other coarse grained permafrost material (Gruber and Hölzle 2008, Scherler et al. 2014). The porosity of

bedrock is assumed to be very low compared to the talus slope and is therefore set to a near-zero value. The so-called air conductivity is an intrinsic permeability which can be understood as a theoretical value of the potential air flow within the material. As Arenson et al. (2006) pointed out, the chosen air conductivity has to be in accordance with the other numerical parameter settings (time step, mesh size) to guarantee numerical stability and convergence. In our study, the air conductivity in the bedrock was set to a near-zero value, assuming that no air circulates within the bedrock. The air conductivity within

the talus and in the air was set to a value of $10^4$ m day$^{-1}$. This value was obtained through numerical test simulations, as in Arenson et al. (2007), who obtained the same value. Higher air conductivity values did not result in numerical convergence and lead to clearly unrealistic results. In comparison to that of the rock material, the heat capacity of the air above the surface is low. Air movement in the atmosphere and its influence on the talus are simplified. There are no turbulent atmospherical fluxes or external wind sources included in the model setup. This is of course a simplification of real-world conditions, but

helps to isolate the process of the internal air circulation within the talus slope without the existence of external forcing by wind.

Finally, the effect of snow is simulated in an indirect way. The thermal effect of snow on the ground is represented through the explicit GST boundary condition at the atmosphere-surface boundary. However, the decoupling of the subsurface from atmospheric circulation regarding the air flow is not represented by this boundary condition. To account for this de-coupling

of air movement, the air conductivity above the surface is parameterised as a function of snow height. Snow height data from the IMIS station Les Attelas (close to the Lapires site, with representative snow data, cf. Staub et al. 2015) was used to parameterise air conductivity. For a snow height below 0.2 m the snow layer does not restrict the circulation. For snow heights between 0.2 m and 0.8 m air conductivity linearly decreases from $10^4$ to 0 m day$^{-1}$ and thus makes the exchange between the air and the talus impossible for snow heights above 0.8 m (cf. Scherler et al. 2013). Morard (2011) has shown

that air aspiration is also possible through the snow, but little is known about the characteristics and importance of this effect.

### 3.3 Boundary and initial conditions

The lower boundary condition is set to a constant value of + 0.6°C. Temperature data from the lowermost thermistor in the Lapires borehole LAP_1108 at 39 m depth show values around + 0.6°C with no seasonal variability (PERMOS 2016a). For the surface-atmosphere boundary condition a 13-year period (2000-2012) of the GST data series from the logger S15 at Lapires (cf. Sect. 2.3, Staub et al. 2015) is used. Initial conditions are set with a constant temperature of -0.2°C in the talus and 0.2°C in the surrounding bedrock, followed by a seven year spin-up using the daily mean values from the S15 GST data as forcing. After seven years a quasi-equilibrium state from the constant initial conditions is reached. After the spin-up procedure the 13-year data series is used at the atmospheric boundary as forcing to show the effects of interannual variability of the driving meteorological variables.

### 3.4 Model experiments

Simulations with four different model setups have been conducted (Table 2). The first setup (CON) allows no air circulation at all and hence only conductive heat transfer is represented, as in most permafrost models and land-surface schemes. This setup serves as a reference simulation. The second setup (CLO) consists of a permanently closed domain which allows air circulation within the talus, but no exchange with the air domain above the surface. In the third setup (OPE) exchange with the modelled air domain above the surface is always possible, whereas the fourth setup (SEA) allows only a seasonal circulation with the air block to represent the seasonal decoupling from talus and atmosphere by the snow cover.

### 4 Results

### 4.1 Air circulation

Figure 3 shows the simulated air circulation and temperature distribution in the full model domain for day 300 (Oct. 27) for the experiment with open boundary conditions between the surface of the talus slope and the atmosphere (OPE). The ascending air flow within the talus slope can be clearly seen, with maximum flow velocities of around 100 m day$^{-1}$, i.e. ~0.001 m s$^{-1}$, which is in the range of the values estimated in other model and observational studies (Tanaka et al. 2000). Meyer et al. (2016) estimated velocities of about 0.07 m s$^{-1}$ for air flow within an ice cave, which can be seen as the upper porosity limit for this kind of intra-talus air circulation. The thermal effect of this circulation is clearly seen with positive temperatures around 2°C in the upper part of the talus slope and negative ones around -1°C in the lower part. Note, that the near-surface layer shows homogeneously negative temperatures at this time of the year due to seasonal freezing from the surface, except for the 10 m around the exit region of the warm ascending air flow in the upper part of the slope, where ground temperatures remain positive.

The air circulation causes aspiration of cold atmospheric air into the lower part of the talus slope, where the lowest temperatures are found (horizontal distance 110 m). This temperature distribution is a clear effect of the induced convective

air circulation. The pure conduction experiment (CON), which served as a reference experiment and only uses conduction for heat transfer, shows a spatially and temporally homogeneous temperature distribution. The small difference between nodes A and B in Fig. 6 is due to the slightly asymmetric model geometry. The spatial variability of the temperature difference to the OPE experiment is shown in Fig. 10 in the supplementary material. Neither negative temperatures nor permafrost conditions are produced in this reference run.

Figure 4 shows the seasonal evolution of the intensity and direction of the air circulation shown in Fig. 3 for the different months. The seasonal reversal of the circulation, as postulated in Delaloye and Lambiel (2005) and Morard et al. (2010, cf. Fig. 1), amongst others, is clearly visible with downslope air flow within the talus slope between June and September and ascending air circulation from November to May. The strongest circulation (snow free and therefore high temperature gradient at the atmosphere – talus boundary) is simulated in July and August, whereas the winter circulation is less strong, but continues over a longer time period (7 months, in contrast to only 3-4 months in summer). Even low temperature gradients cause a circulation in the modelling results. However, a weak circulation has almost no thermal effect and thus conduction dominates.

Figure 5 shows the differences of the simulated seasonal circulation depicted in Fig. 4 between the three experiments CLO, OPE and SEA. The seasons are shown as 3-month averages regarding their typical conditions at high elevation, i.e.: winter – JFM; spring – AMJ; summer – JAS; autumn – OND. The transition period with reduced air circulation between April and June is seen in all experiments, however, it has to be noted that during these 3 months the circulation reverses (cf. Fig. 4) and downslope and upslope air flow will contribute to near-zero air flow in a 3-month average. The strong downslope air flow in summer is seen in OPE and SEA, but much less so in CLO, as only little space in the near-surface part of the talus slope is available for the upward-directed backflow of air. In CLO the circulation is also weaker in winter, which clearly illustrates that an open boundary between atmosphere and talus slope is necessary to produce stronger air circulation. However, it can be seen that in CLO a seasonally reversing air circulation also develops.

The OPE and SEA experiments differ regarding their open and seasonally closed boundary condition, the latter resembling the insulating effects of the seasonal snow cover. The comparison shows that between July and December almost no differences can be seen, as the snow cover is absent or not yet thick enough to dampen the air exchange between talus slope and atmosphere. On the contrary, strong damping effects can be observed between January and June, due to the presence of an up to 2 m thick snow cover (cf. Fig. 8).

## 4.2 Ground temperature

The effect of the seasonally reversible air circulation on the spatial ground temperature distribution was already shown in Fig. 3. Figure 6 shows the seasonal and interannual variations of this temperature effect for all four experiments. Temperature curves in the upper part of the talus slope (Node A) indicate that temperatures are always positive with a seasonal variability of around 1°C. An exception is summer 2003, where the effect of the exceptional heat wave (cf. Schär et al. 2004, Hilbich et al. 2008) is clearly shown by a temperature increase between 1°C (CON) and 3°C (OPE and SEA). Apart

from this anomalous year, interannual variability is small in the upper part of the talus slope. In addition, the effect of the snow cover (SEA A compared to OPE A) is small with no differences in summer and maximal temperature differences of around 0.2°C in winter.

In contrast, the interannual variability in the lower part of the talus slope is much more pronounced. Here, temperatures in the OPE and SEA experiments show permafrost conditions which are almost conserved during the summer 2003 heat wave. Temperature decreases exceeding 3°C can be observed during the cold winters 2004/05 and 2005/06. After that, temperatures increase again steadily, but still show permafrost conditions at the end of the simulation period in 2012 for the OPE experiment.

The mean temperatures in the lower part of the talus at node B over the 13 year modelling period decrease by 0.28 °C (CLO), 0.94°C (SEA) and 1.19°C (OPE), respectively, compared to the CON experiment without convective cooling. Temperature differences between the upper and lower parts of the talus slope are up to 6°C during the warm summer 2003 and the cold winters in 2004/05 and 2005/06, which is an effect of the air circulation alone, as no significant temperature differences were obtained in the conductive reference experiment (CON, grey and black lines on Fig. 6). Differences between the OPE and SEA experiments are around 0.2°C for the lower part of the talus slope, demonstrating the critical role of the snow cover in damping the cooling effect of the talus slope circulation.

## 5 Discussion

The simulations qualitatively reproduce the internal air circulation observed in low- and high elevation talus slopes and they are consistent with many different observations. For a critical analysis of the reliability of the model approach and its results, the degree of practicality of the processes in the model, its sensitivity to model parameters, a comparison with observed borehole temperatures in the Lapires talus slope as well as a discussion of the strengths and weaknesses of the model will be addressed below.

### 5.1 Process analysis

Figure 7 shows the velocity of the air flow within the talus slope as a function of driving GST data. It is clear that there is a strong correlation between high positive GST data and strong downslope (positive) air flow and correspondingly, negative GST data with upslope (negative) air flow. This is in good agreement with the underlying process, where a high temperature gradient between the inside and outside of the talus slope leads to ascending air if the interior is warmer and descending air if the interior is colder than the outside air (Fig. 1). As the temperature within the talus slope is close to 0°C, a GST of 0°C corresponds roughly to zero air displacement, as there is no temperature gradient. The high correlation coefficient ($r^2$) in the OPE and the SEA experiments shows that GST is a very important factor in explaining the air flow through the talus. The slightly lower $r^2$ in the CLO experiment is partly due to the weaker air circulation and thus greater influence of the values near the zero curtain, but may also indicate that the closed model conditions are less representative.

Figure 8 shows an analysis of the simulated air flow in dependence of the presence/absence of a snow cover for the three different experiments. As shown in the previous figures, the velocities are lowest for the CLO experiment, and the air flows downslope in summer and upslope in winter in all experiments. In addition, velocities in summer for OPE and SEA show a similar behaviour, as no snow cover is present. Due to the more efficient ground cooling in winter, the OPE experiment shows slightly higher air velocity values. However, large differences can be seen in late autumn and during winter. Clearly, winter velocities are highest for OPE, as there is no snow cover inhibiting the coupling between talus slope and atmosphere in this experiment. In contrast, maximal velocities for the ascending air flow are obtained in late autumn in the SEA experiment, which can be seen as the most realistic simulation, as it includes the non-linear damping effect of the snow cover. In late autumn, air (and also ground surface) temperatures can be negative due to strong radiative cooling at the surface at night. In the absence of a thick snow cover (e.g., in autumn/early winter 2005, Fig. 8) a strong air circulation can develop, as the temperature gradient between interior and outside is high. As soon as the snow cover completely insulates the ground, the air circulation is severely damped as can be seen by the increasing and then flattening velocity curve in the SEA experiment. A similar, but much weaker effect can be seen in early summer when the snow cover thickness decreases again. However, as the temperature gradient is much weaker during the main snow melt period, its effect on the evolution of the air circulation is also much smaller.

## 5.2 Model sensitivity analysis

Sensitivity simulations with different slope angles showed a strong influence of the slope angle on the intensity of the air circulation and therefore on temperature evolution. Not surprisingly, higher slope angles lead to higher velocities and larger temperature contrasts due to the stronger gravitational forces, as was also noted in previous studies (e.g., Guodong et al. 2007). As the mesh geometry (cf. Fig. 2) changes with changing slope angles, the obtained temperature dependencies on slope angle cannot be compared quantitatively. In addition, the circulation pattern changes with increasing slope angle. For the simulations with high slope angles (such as the 33° depicted in Fig. 2, which was used in all simulations shown above) only one 2-dimensional advection cell is present, with seasonal reversal of the flow direction (cf. Figs. 3-5). With decreasing slope angle this circulation cell decreases in strength and additionaly 1-dimensional vertical convection cells are formed, which do not extend over the full model domain (not shown). For slope angles lower than around 5° the air flow is dominated by several individual small vertical convection cells.

Guodong et al. (2007) describe a similar relation between slope angle and circulation patterns in laboratory experiments. They found a minimal slope angle of 15° for connected advection cells and a dependence on the Rayleigh number for higher slope angles (cf. also section 2.1). In practice and in our model simulations it is difficult to distinguish clearly between horizontal advective and vertical convection cells as there is usually a gradual transition between the two phenomena.. Using in-situ measurements, Hanson and Hölzle (2004) showed that for the case of rock glacier Murtèl, a comparatively flat rock glacier with large furrows and ridges inhibiting the lateral air circulation, the vertical exchange of air is more important than

advective processes. A thorough observation- and simulation-based analysis of these dependencies still needs to be carried out.

In addition, the model is very sensitive regarding the so-called air conductivity parameter introduced above. Arenson et al. (2006) concluded that finding a physically-consistent combination of air conductivity, and temporal and spatial discretisation is the biggest practical problem affecting numerical stability in soil temperature simulations. In our simulations only a narrow range of air conductivities allow for convergent calculations that produce convective circulation patterns for a given mesh and time step. Similar results were obtained in GeoStudio (2013a) and Arenson et al. (2007), which suggests that the air circulation should be used as a calibration parameter wherever subsurface temperature data are available, especially as there are generally no in-situ air velocity measurements within the talus slopes.

## 5.3 Comparison of modelled and measured ground temperature data

Finally, the idealised experiments discussed above can be compared to the observed temperature evolution in a borehole situated in the central part of the Lapires talus slope. Even though the driving GST values were taken from this talus slope, this comparison can only be qualitative, as no temperature calibration has been attempted in our model and there are several differences between our model setup and the real conditions. The most notable difference is the presence of a massive ice core at Lapires (Hilbich 2010, Scapozza et al. 2015), which was not included in our simulations. Although technical problems prevent a detailed analysis of the borehole temperatures until 2010, permafrost conditions are clearly present with an active layer thickness ranging between 4-6 metres (PERMOS 2016a). Generally, the simulated values are similar to the observed temperatures, however, a small cold bias can be identified in the model (cf. Fig. 9). As a result, the modelled active layer is thinner and the penetration depth of the winter cooling is greater than in the observations.

This may have several reasons. Firstly, the mean of the driving GST values of the logger LAP_S15 is colder than the mean of the uppermost temperature sensor in the borehole (Staub et al. 2015). Secondly, the massive ice core influences the ground thermal regime regarding (i) the induced air circulation patterns by blocking the air flow in a central region of the talus between 4 m and approximately 15 m depth, (ii) its different thermal properties from the air/rock material of the simulated talus slope and (iii) the latent heat necessary for phase changes. Finally, due to numerical effectivity, the depth to the bedrock was much lower in the model than in reality (Scapozza et al. 2015), which may have an additional effect on the observed discrepancies.

## 5.4 Model strengths and weaknesses

Modelling air circulation and thus the convective heat transfer within a talus slope is an important contribution to the understanding of the complex thermal regime of a talus slope in mountainous environments. The strength of this modelling approach lies in the fact that convective heat transfer is explicitly modelled in two dimensions, and it is a first step to an explicit description of this process in more sophisticated long term permafrost models where ventilation has so far been neglected or only parameterized (Marmy et al. 2016, Luethi et al. 2016). Weaknesses mainly concern the lack of latent heat

effects caused by melting/freezing of ice. The model domain in this study is assumed to be dry and hence there is no ice formation. This allows assessment the influence of the convective heat transfer by air flow in an isolated way but neglects the interaction of water, ice and air. In ice-rich talus slopes this interaction may have an important role but was beyond the scope of this study and should be part of future model improvements. Previous studies on the effect of convective heat

transfer in porous media in permafrost environments also neglect the influence of water flow and assume the numeric domain to be dry (Goering and Kumar 1996, Arenson et al. 2006, 2007, Guodong et al. 2007). The representation of snow is still quite poor. The interactions between the talus slope circulation and snow layer are complex and not yet fully understood. Melt holes due to warm air exits are frequently observed at the top of a talus slope (Morard 2011) and probably have a significant impact on the air flow path and the air velocity. The pronounced ground cooling in the lower part may also

influence the snow cover and some refreezing of percolating melt water may take place. Finally, the air block above the talus slope is a very simplified representation of the atmosphere. First of all, a pressure boundary condition could have been applied to link the talus slope to the atmosphere. Air flow at the boundary is not known and would have needed further parameterizations - so this approach was dismissed. Secondly, we assume that the high thermal conductivity of air used in this study to some extent compensates the missing turbulent fluxes which are prevented by the actual low air conductivity.

This may affect the absolute values but allows numerically consistent simulations which represent the underlying process within the talus slope. Modelling the atmospheric air flow explicitly, for example, the effect of wind speed on the intra-talus circulation, would, however, strongly increase the complexity of the model.

## 6 Conclusions and outlook

A numerical study of convective heat transfer by air flow in a 2-dimensional talus slope has been conducted. The results

show that without forced convection a seasonally alternating circulation develops which leads to a convective heat transfer within the talus and thus to a cooling effect in the lower part of the talus slope. It therefore has a significant influence on the thermal regime in a talus slope. Sensitivity studies with a conduction model show that this cooling can be crucial to maintain the frozen state of the ground during the warm season under warming climate conditions. Furthermore, sensitivity studies, using different atmospheric boundary conditions, were conducted by simulating a closed-to-atmosphere boundary, and open-

to-atmosphere boundary and a seasonally closed-to-atmosphere boundary to represent the seasonal decoupling by a snow cover. The results show that an open-to-atmosphere boundary leads to the most efficient cooling. Compared to a conduction only setup, a cooling of 0.28 °C (CLO), 0.94°C (SEA) and 1.19°C (OPE) , respectively, was found in the lower part of the talus slope as mean value over the whole 13 years. Further sensitivity studies showed the dependence of strong cooling effects on a high air velocity in the ground as well as  on an increase in slope angle. However, the circulation pattern does

not change significantly in these cases.

So far, this phenomenon has mainly been documented by field studies (e.g., Delaloye et al. 2003, Gude et al. 2003, Sawada et al. 2005) and the explicit modelling of this process now confirms their findings. For simplicity, we neglected the

formation of ice, which may be a reasonable assumption for low-elevation, but not for high alpine talus slopes (Delaloye and Lambiel 2005, Morard et al. 2010, Hilbich 2010, Scapozza et al. 2015). Furthermore the infiltration of melt water and precipitation as well as intra talus water flows (Luethi et al. 2016), which can lead to advective heat transfer and thus have an influence on the ground thermal regime, are neglected. This will have to be improved in a future study. Future work should

therefore aim to model the internal air circulation for a specific site taking into account a more detailed structure like an impermeable ice core to get a better validation and site specific quantifications of the process. Furthermore efforts have to be made to understand the complex coupling of snow and air circulation within a talus slope.

*Data availability.* The borehole temperature data set of the PERMOS network is published under PERMOS (2016a). The

data are available at doi:10.13093/permos-2016-01. The ground surface temperature (GST) data can be obtained on request from PERMOS network (www.permos.ch). The model output is stored at the Department of Geosciences and can be obtained through the corresponding author.

*Competing interests.* The authors declare that they have no conflict of interest.

*Acknowledgements.* This study was conducted within the SNF-Sinergia project TEMPS financed by the Swiss National Science Foundation (project n° CRSII2 136279) and the authors would like to thank all colleagues within the project for their valuable input during meetings and conferences. We would like to thank Dr. B. Staub and Prof. R. Delaloye for their valuable input regarding the Lapires talus slope. Data of the Lapires field site were thankfully provided by the PERMOS

network (Permafrost Monitoring Switzerland) and snow data for the "Les Attelas" from the IMIS network (Intercantonal Measurement and Information System). Furthermore we would like especially to thank Dr. L. Arenson for sharing his experience with GeoStudio and his valuable input as reviewer, an anonymous reviewer and the editor Marcia Phillips for their reviews and helpful suggestions.

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

**Table 1: Material properties used for the GeoStudio simulations in this study (based on Schneider 2014, Arenson and Sego 2006, Arenson et al. 2007 and Scapozza et al. 2015, see text). The snow layer only acts as an idealised boundary that seasonally damps and prohibits air exchange between talus and atmosphere through coupling of the air conductivity value to the snow height. See text for further explanations.**

| material | thermal conductivity | | specific heat capacity | vol. heat capacity | porosity | air conductivity |
|---|---|---|---|---|---|---|
| | $[W\ K^{-1}m^{-1}]$ | $[kJ\ day^{-1}\ m^{-1}\ K^{-1}]$ | $[J\ kg^{-1}\ K^{-1}]$ | $[kJ\ m^{-3}\ K^{-1}]$ | | $[m\ day^{-1}]$ |
| bedrock | 2.4 | 207 | 99 | 2500 | 0.01 | 0.01 |
| talus | 1.2 | 103 | 904 | 1256 | 0.5 | $10^4$ |
| air | 1.2 | 103 | 1 | 1.3 | 1 | $10^4$ |
| snow | 1.2 | 129 | 1 | 1.3 | 1 | $0-10^4$ |

**Table 2: List and description of model experiments conducted.**

| Name | Processes included | Description |
|---|---|---|
| **Conduction only (CON)** | Conduction only | - |
| **Closed (CLO)** | Conduction and convection | No exchange with the air block |
| **Open (OPE)** | Conduction and convection | Unrestricted exchange with the air block |
| **Seasonally closed (SEA)** | Conduction and convection | Seasonally restricted exchange with the air block |

**List of Figures:**

Figure 1: Schematic view of a seasonally alternating air circulation within a talus slope (modified from Morard et al. 2010). $T_{ao}$ and $T_{ai}$ denote air temperature outside and inside the blocky material, respectively.

Figure 2: Model setup and mesh showing the four sub-domains used in the talus slope experiments. The points (nodes) marked by A and B are reference points which are further analysed below.

Figure 3: Simulated temperature distribution (colours) and air current vectors in the talus slope for the open experiment (OPE) for day 300 (winter circulation). The intensity of the circulation is marked by the grey/black colour of the vector arrows and is given in m day$^{-1}$.

Figure 4: Mean monthly air flow of the open experiment (OPE) over the 13 modelled years. The dashed line marks the domain of the talus slope. The seasonal differences in intensity and direction of the air circulation, are clearly shown by the direction and colour of the arrows.

Figure 5: Modelled seasonal mean air flow for the modelled 13 years for the different atmospheric boundary conditions: (top) closed to atmosphere (CLO), (middle) open to atmosphere (OPE) and (bottom) seasonally closed to atmosphere (SEA). The dashed line is the talus slope.

Figure 6: Temperature evolution at two different locations in the upper (node A, solid lines, see Fig. 2) and lower (node B, dashed lines, see Fig. 2) part of the talus slope for the CLO (red), the OPE (blue), the SEA (purple) and the CON (black) experiment.

Figure 7: Relation between velocity of the air flow at node B (see Fig. 2) and the forcing ground surface temperature (GST) for the three different experiments OPE, SEA and CLO. Positive velocities indicate downslope movement and negative velocities indicate upslope movement of air.

Figure 8: Velocity of the simulated air flow at node B (see Fig. 2) in the experiments OPE, SEA and CLO for the period 2010-2013 in relation to the presence of a snow cover (grey bars). Positive (negative) velocities correspond to downslope (upslope) air flow.

Figure 9: (a) Temperature plot along a perpendicular profile through node B for the OPE simulation for the 13 year period. (b) Borehole temperature from borehole LAP_0198 at the Lapires field site (PERMOS 2016a). Missing values due to technical problems at greater depths are plotted in dark grey.

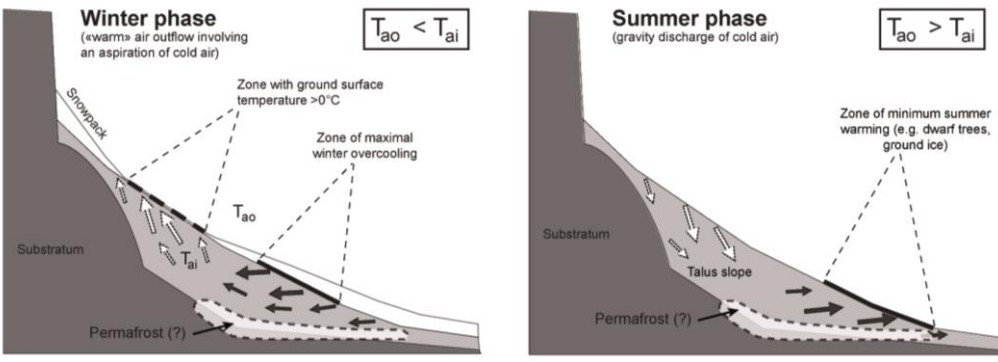

**Figure 1: Schematic view of a seasonally alternating air circulation within a talus slope (modified from Morard et al. 2010).** $T_{ao}$ and $T_{ai}$ denote air temperature outside and inside the blocky material, respectively.

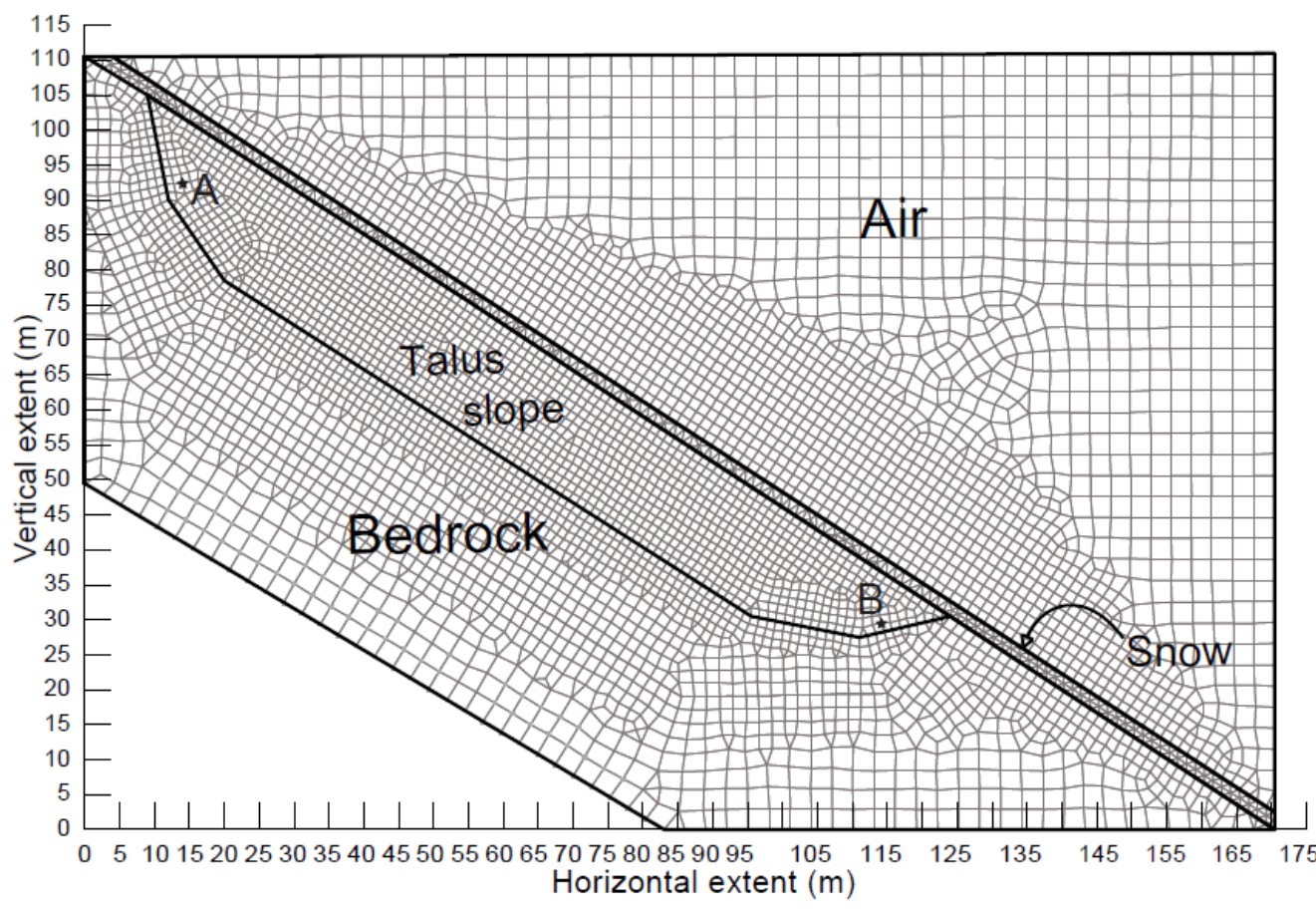

**Figure 2: Model setup and mesh showing the four sub-domains used in the talus slope experiments. The points (nodes) marked by A and B are reference points which are further analysed below.**

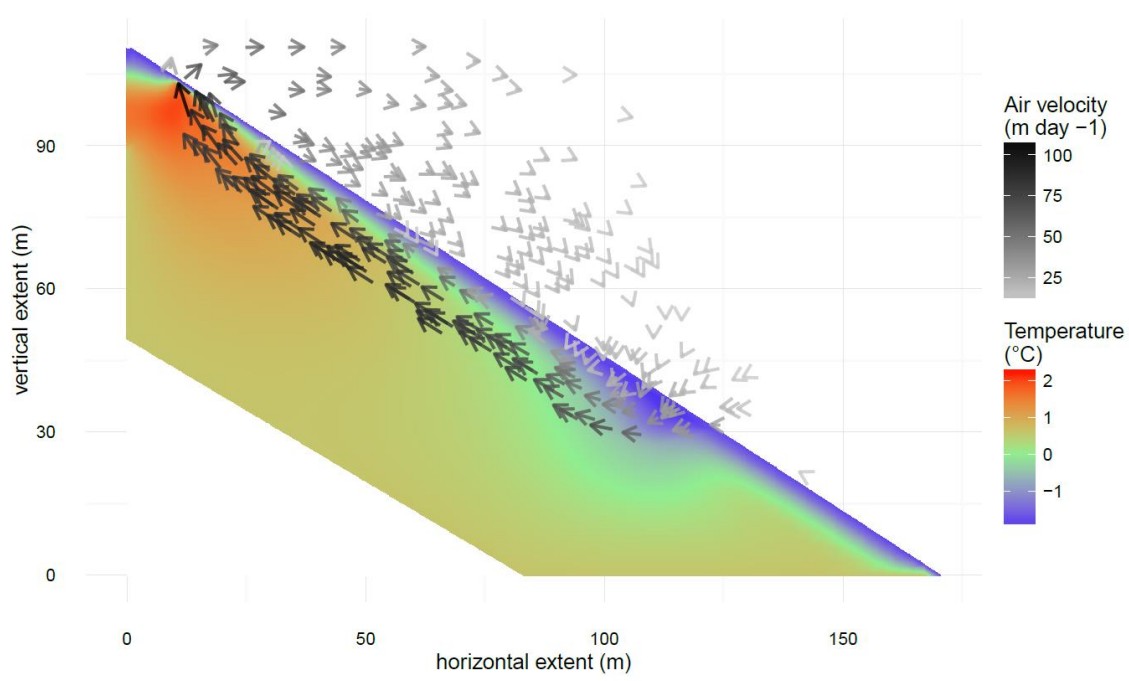

**Figure 3: Simulated temperature distribution (colours) and air current vectors in the talus slope for the open experiment (OPE) for day 300 (winter circulation). The intensity of the circulation is marked by the grey/black colour of the vector arrows and is given in m day$^{-1}$.**

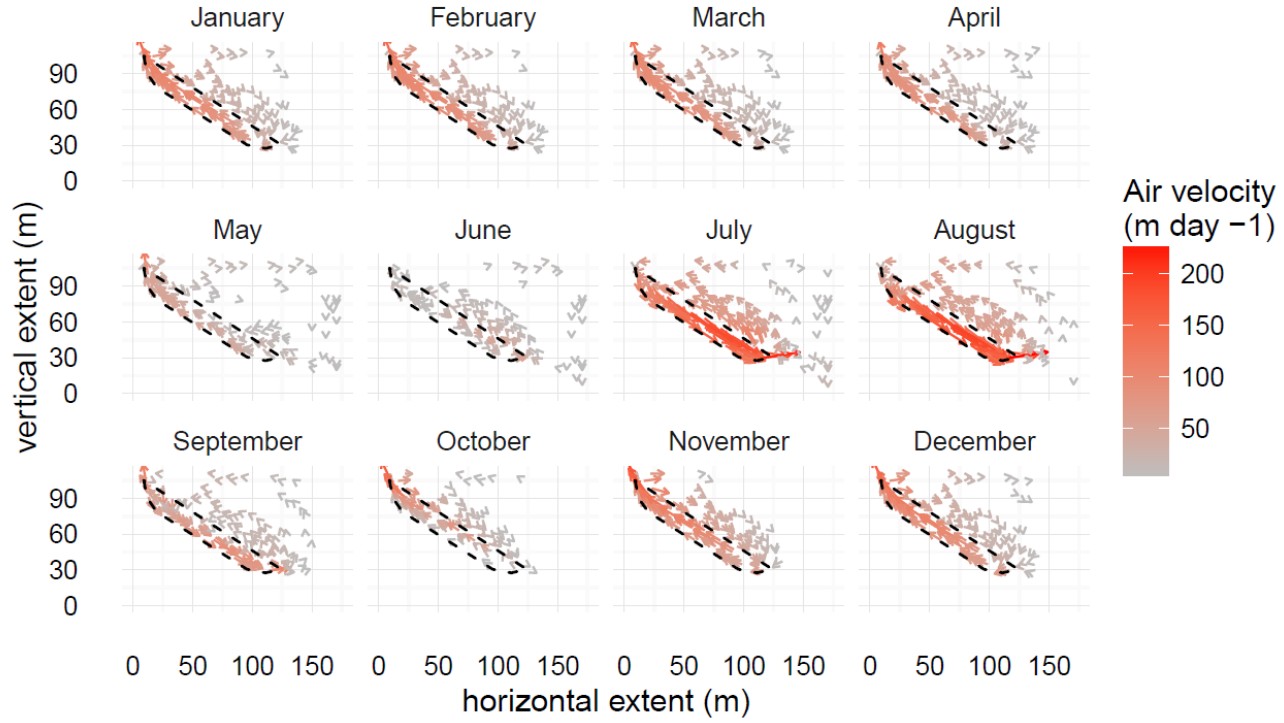

**Figure 4: Mean monthly air flow of the open experiment (OPE) over the 13 modelled years. The dashed line marks the domain of the talus slope. The seasonal differences in intensity and direction of the air circulation, are clearly shown by the direction and colour of the arrows.**

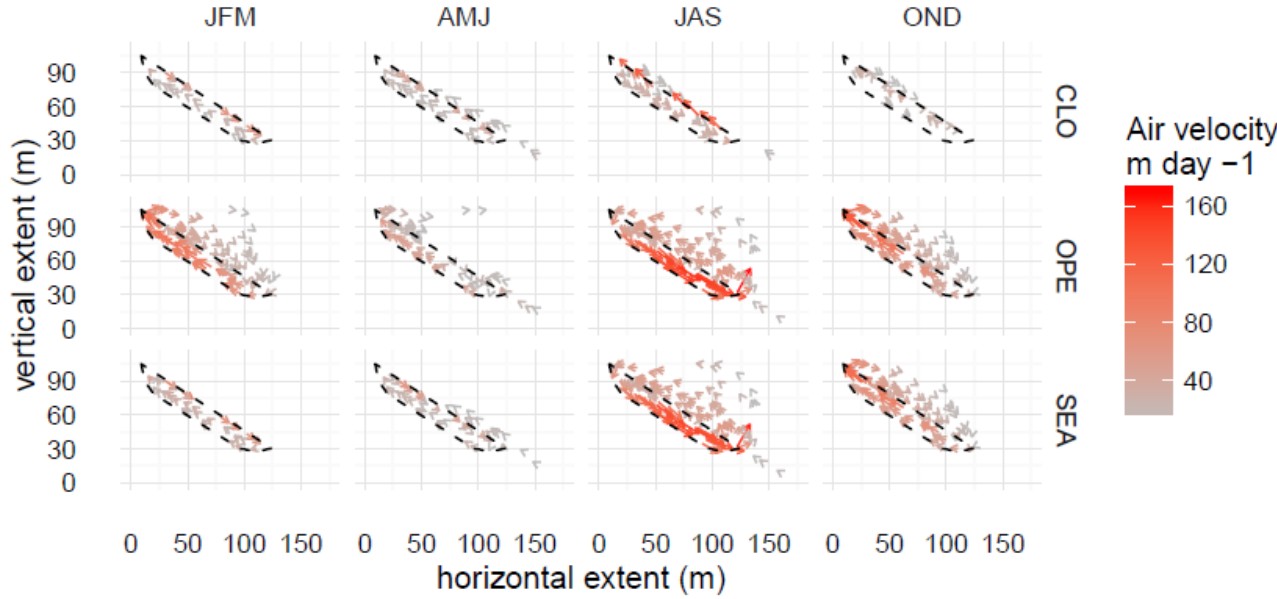

**Figure 5: Modelled seasonal mean air flow for the modelled 13 years for the different atmospheric boundary conditions: (top) closed to atmosphere (CLO), (middle) open to atmosphere (OPE) and (bottom) seasonally closed to atmosphere (SEA). The dashed line is the talus slope.**

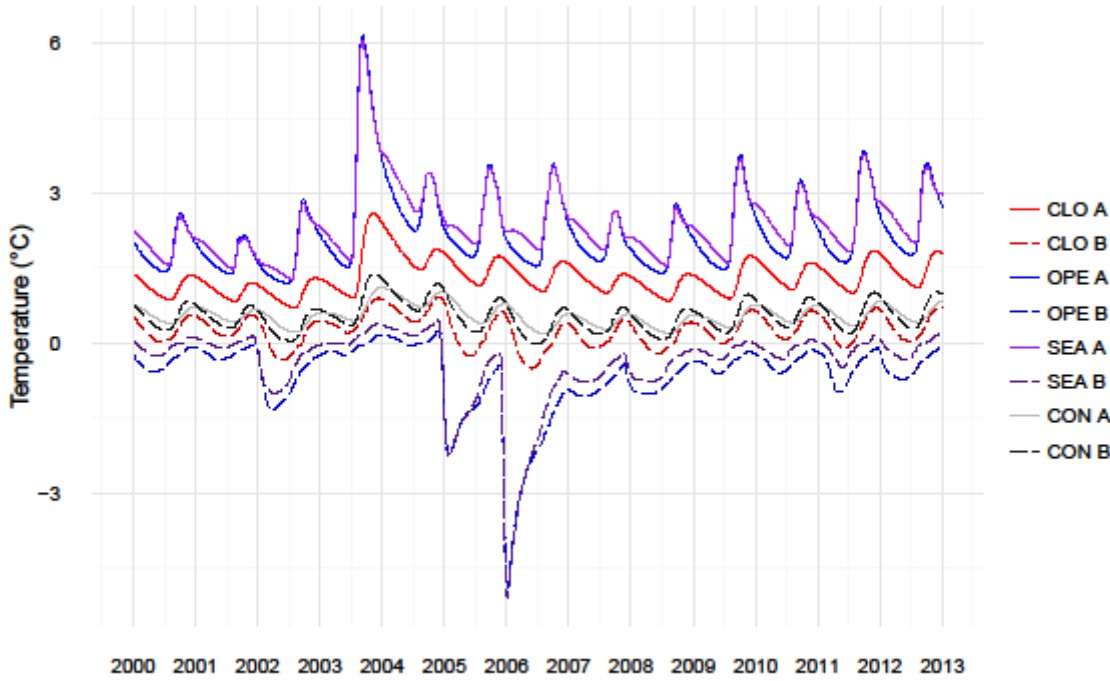

Figure 6: Temperature evolution at two different locations in the upper (node A, solid lines, see Fig. 2) and lower (node B, dashed lines, see Fig. 2) part of the talus slope for the CLO (red), the OPE (blue), the SEA (purple) and the CON (black) experiment.

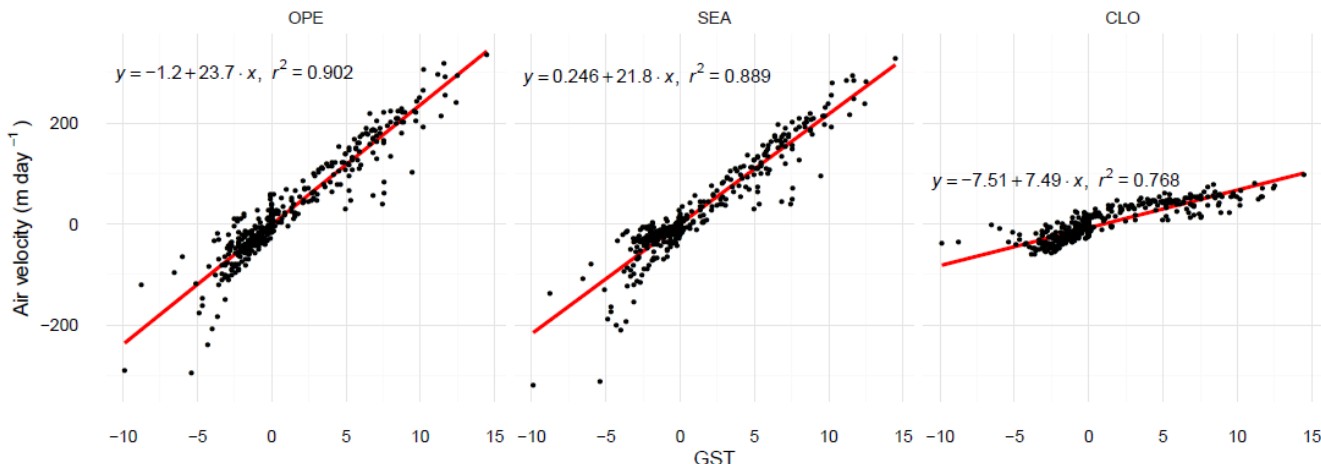

**Figure 7: Relation between velocity of the air flow at node B (see Fig. 2) and the forcing ground surface temperature (GST) for the three different experiments OPE, SEA and CLO. Positive velocities indicate downslope movement and negative velocities indicate upslope movement of air.**

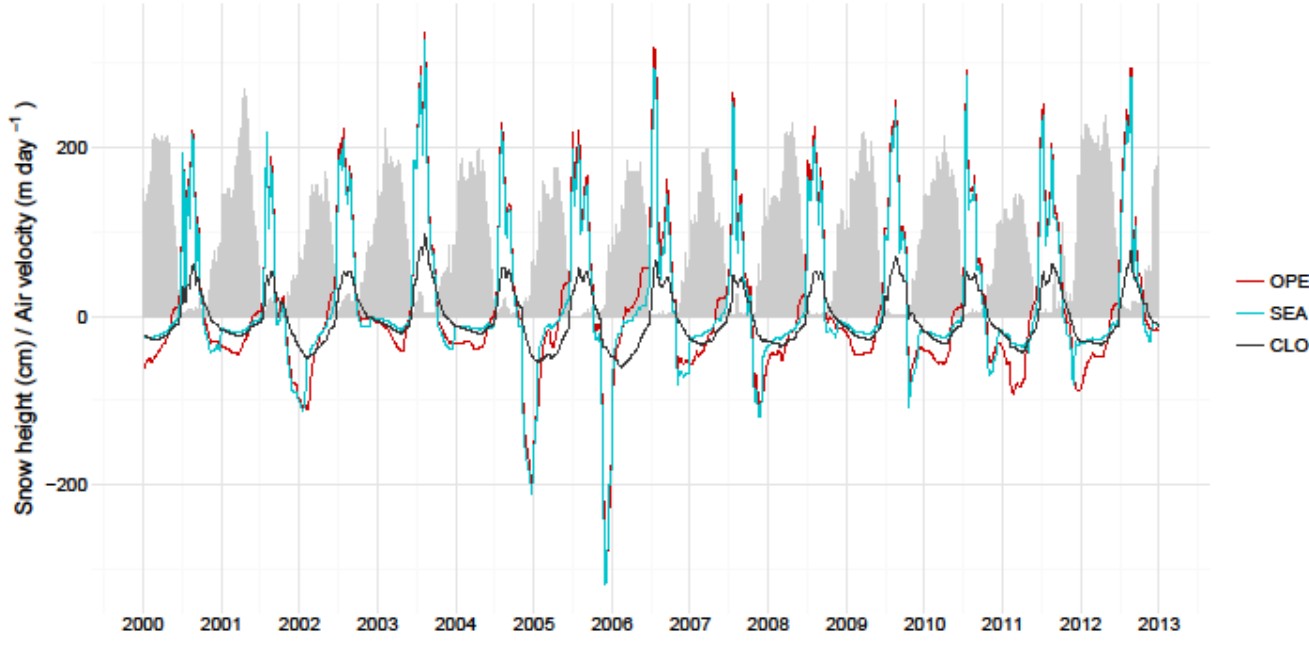

**Figure 8: Velocity of the simulated air flow at node B (see Fig. 2) in the experiments OPE, SEA and CLO for the period 2010-2013 in relation to the presence of a snow cover (grey bars). Positive (negative) velocities correspond to downslope (upslope) air flow.**

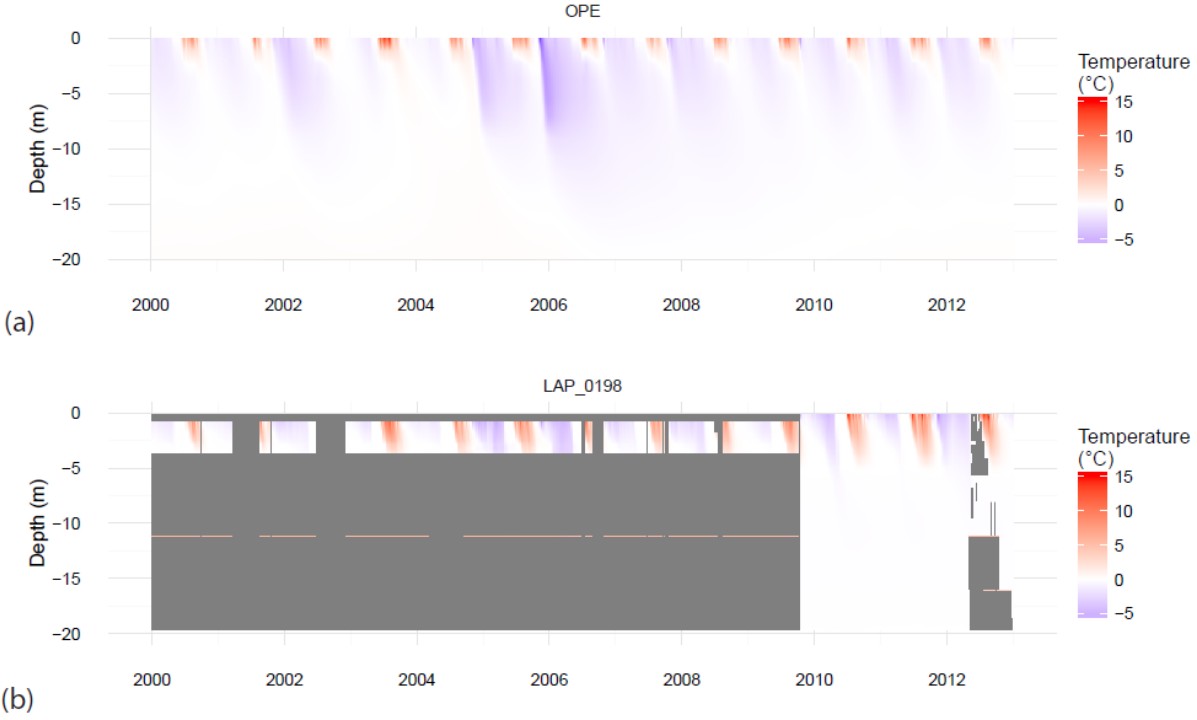

**Figure 9: (a) Temperature plot along a perpendicular profile through node B for the OPE simulation for the 13 year period. (b) Borehole temperature from borehole LAP_0198 at the Lapires field site (PERMOS 2016a). Missing values due to technical problems at greater depths are plotted in dark grey.**

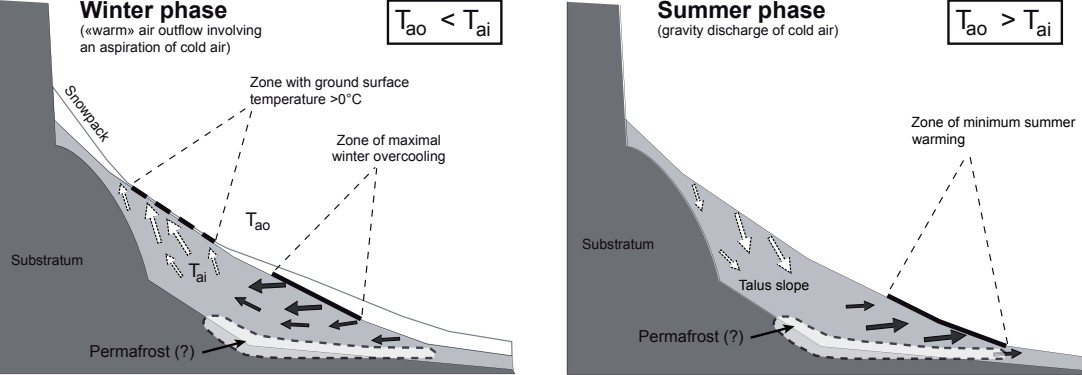

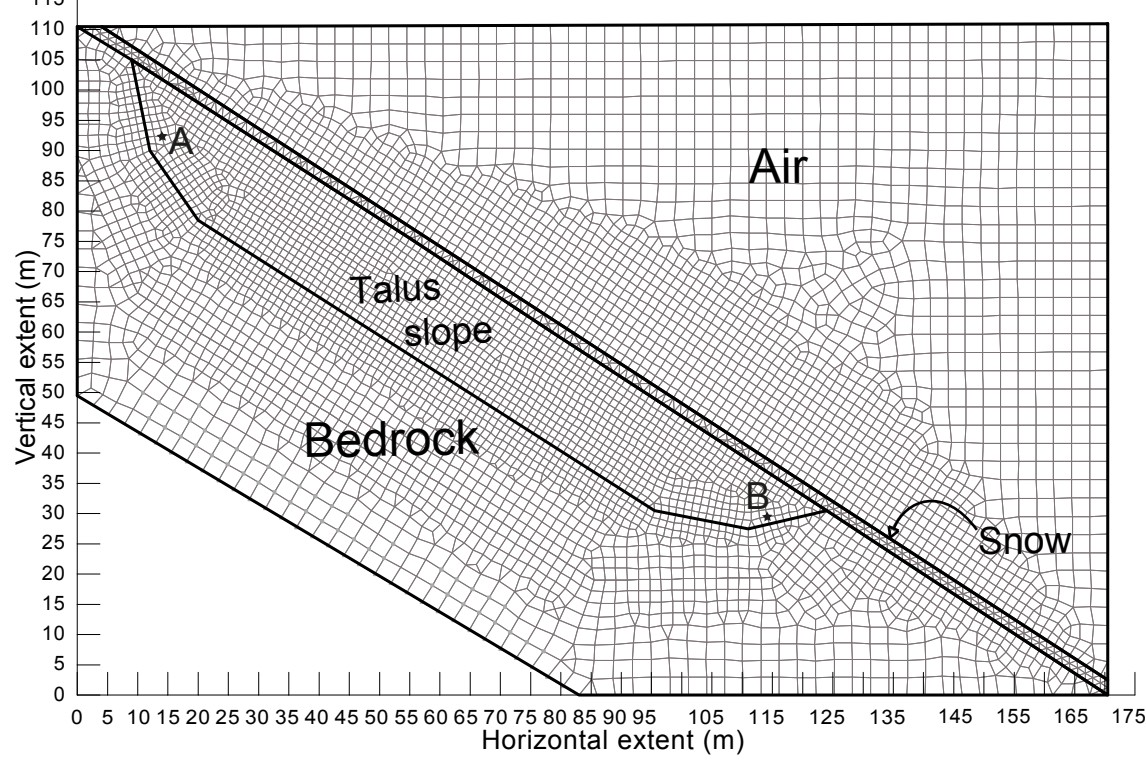

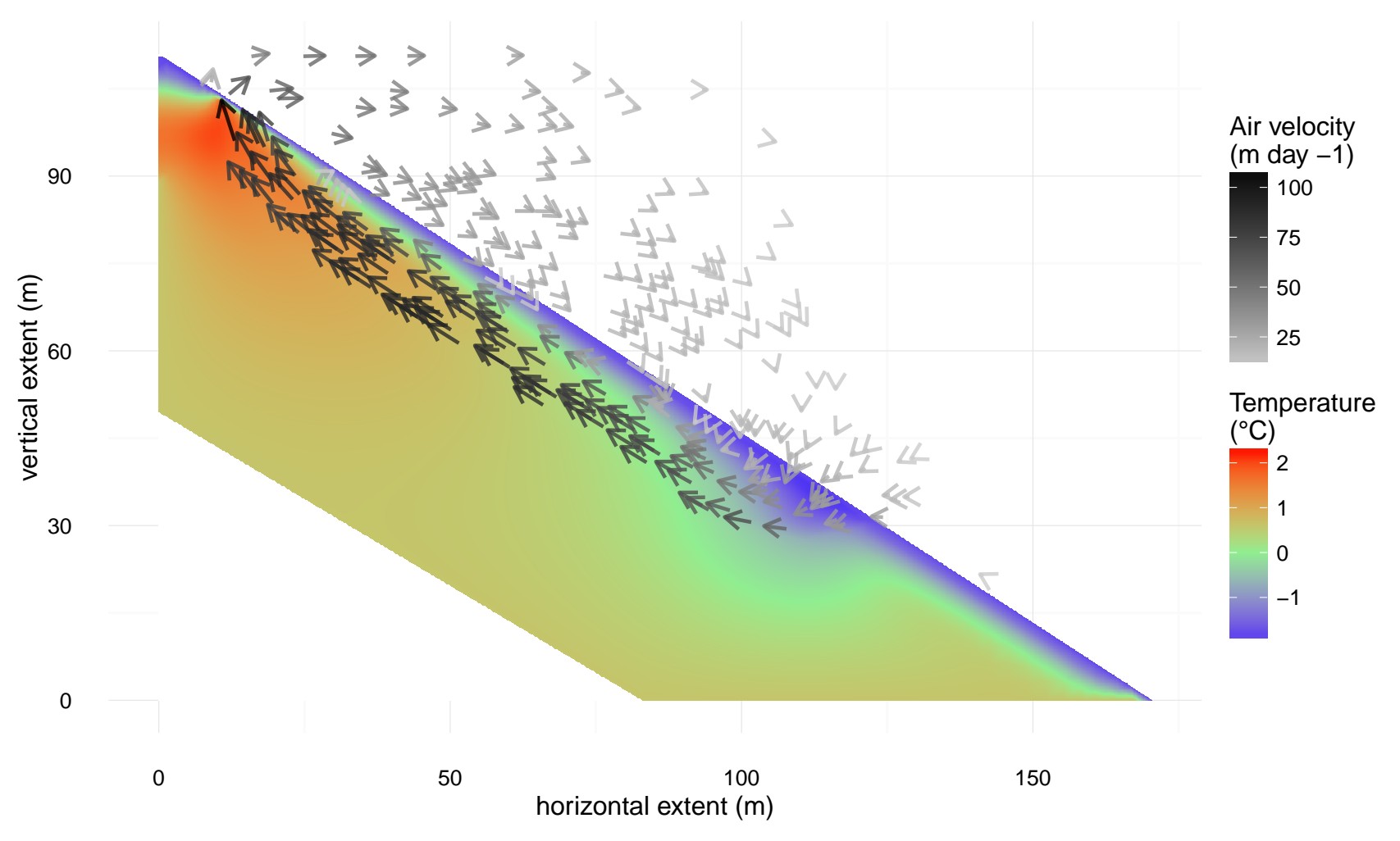

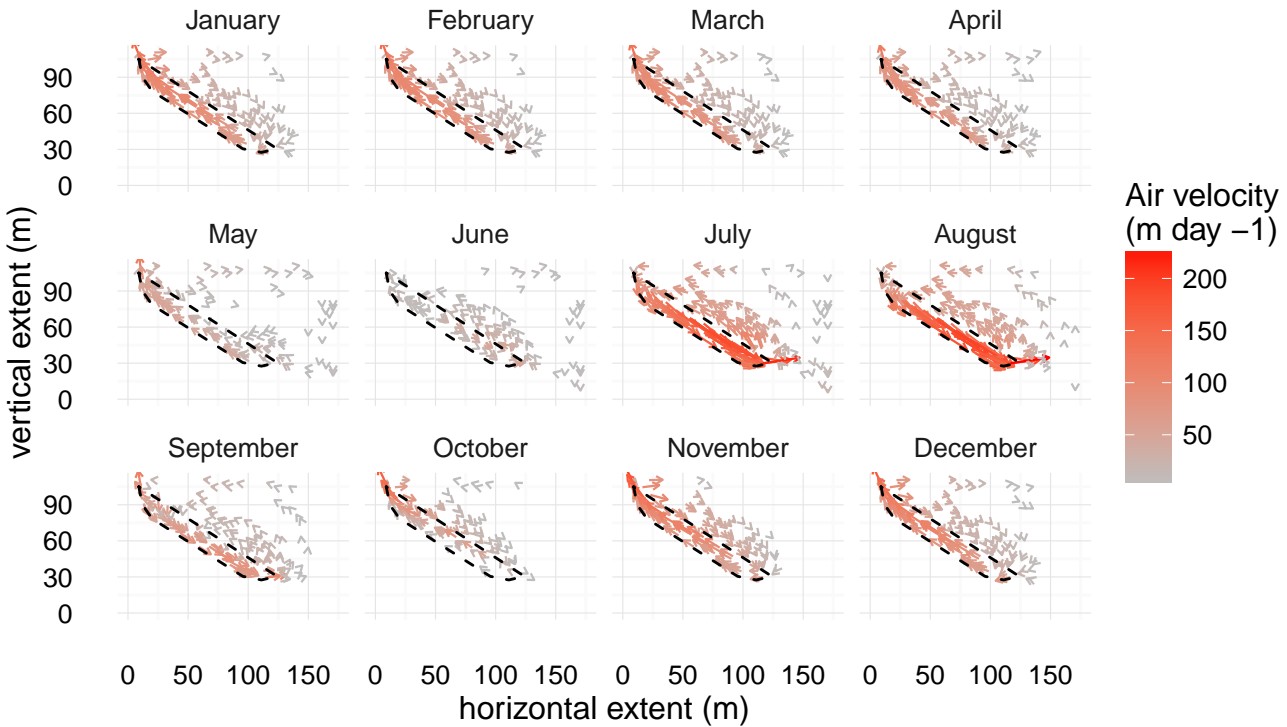

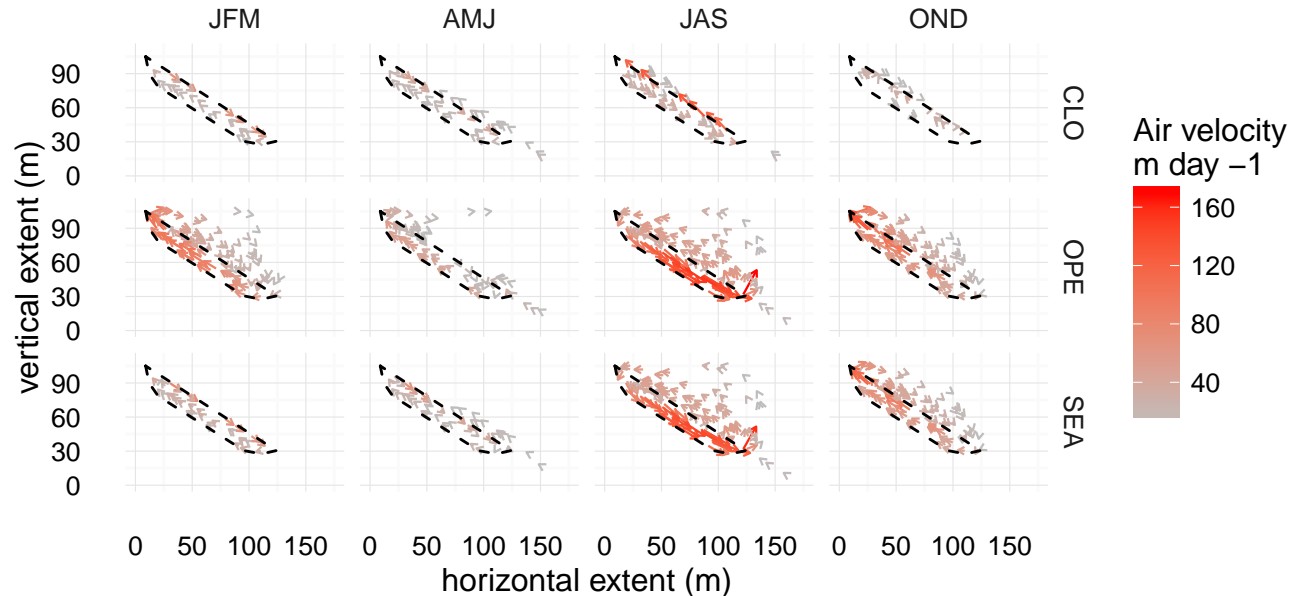

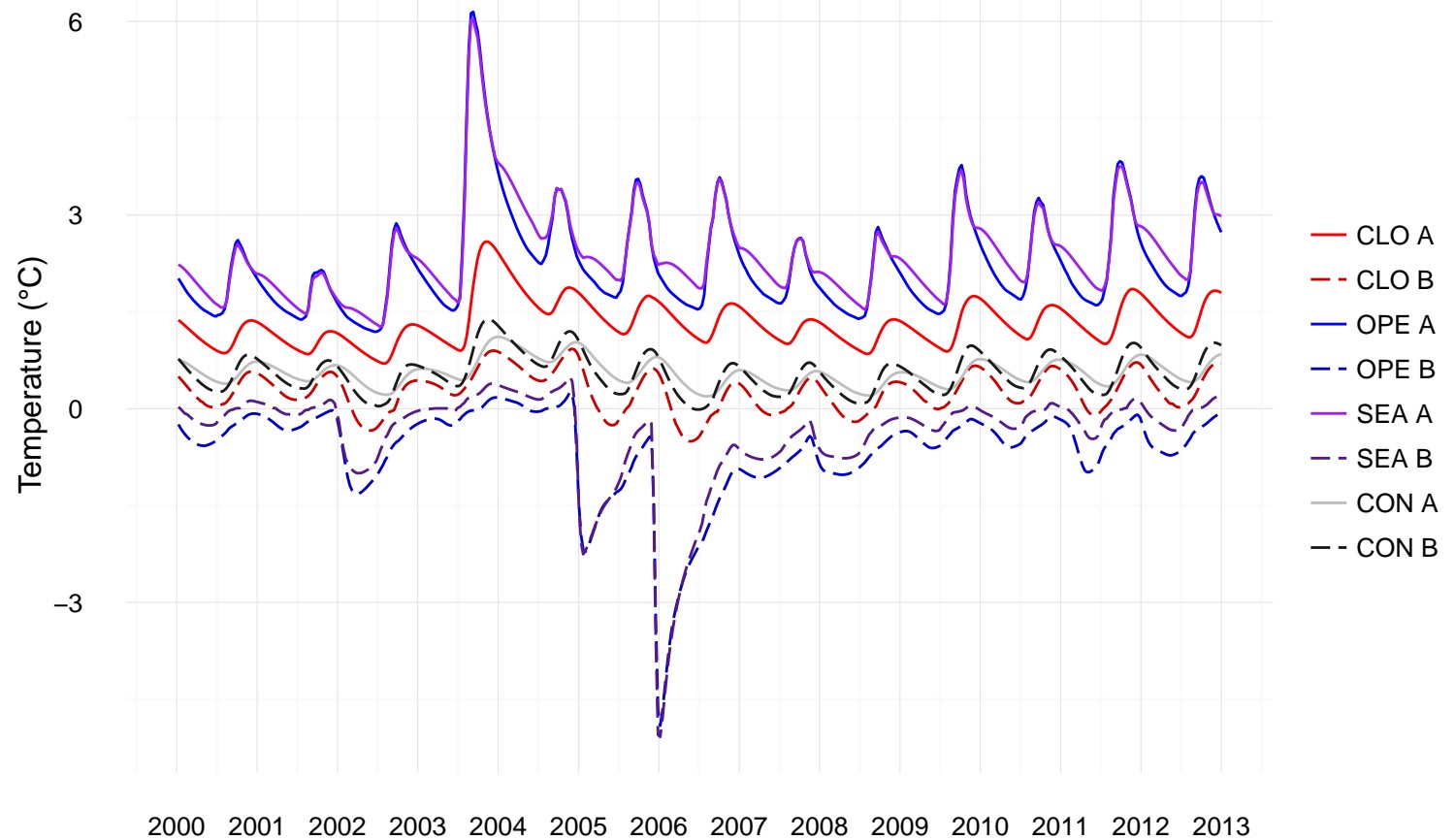

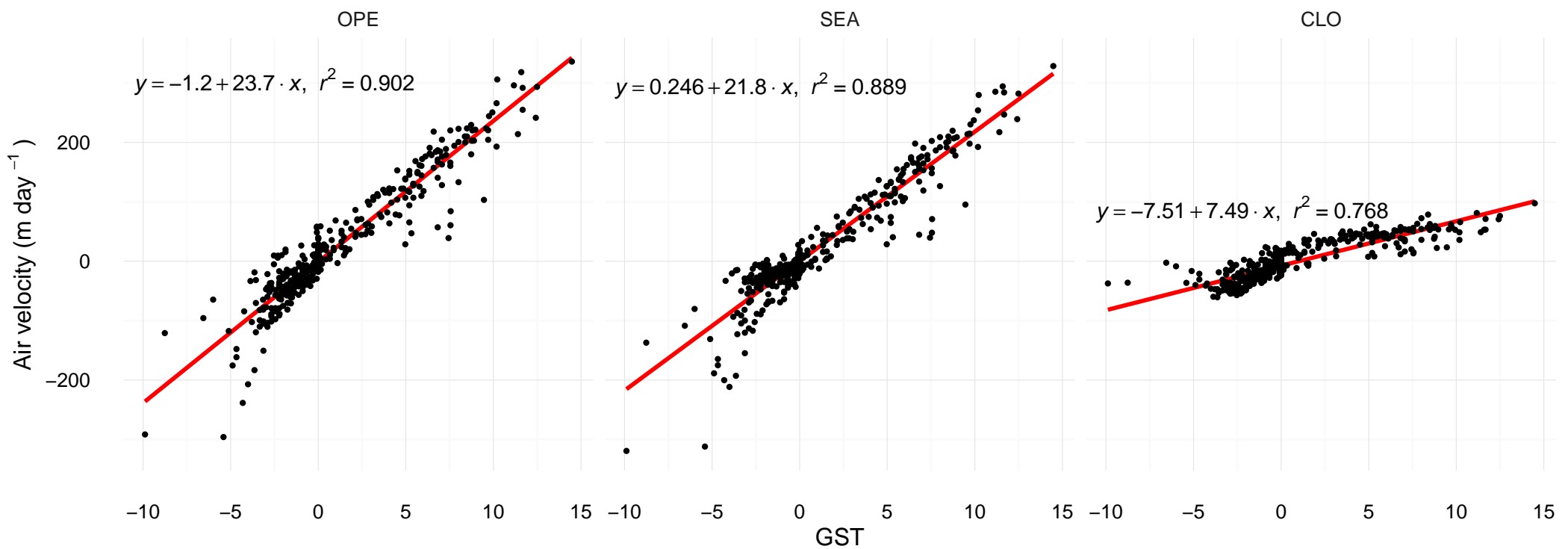

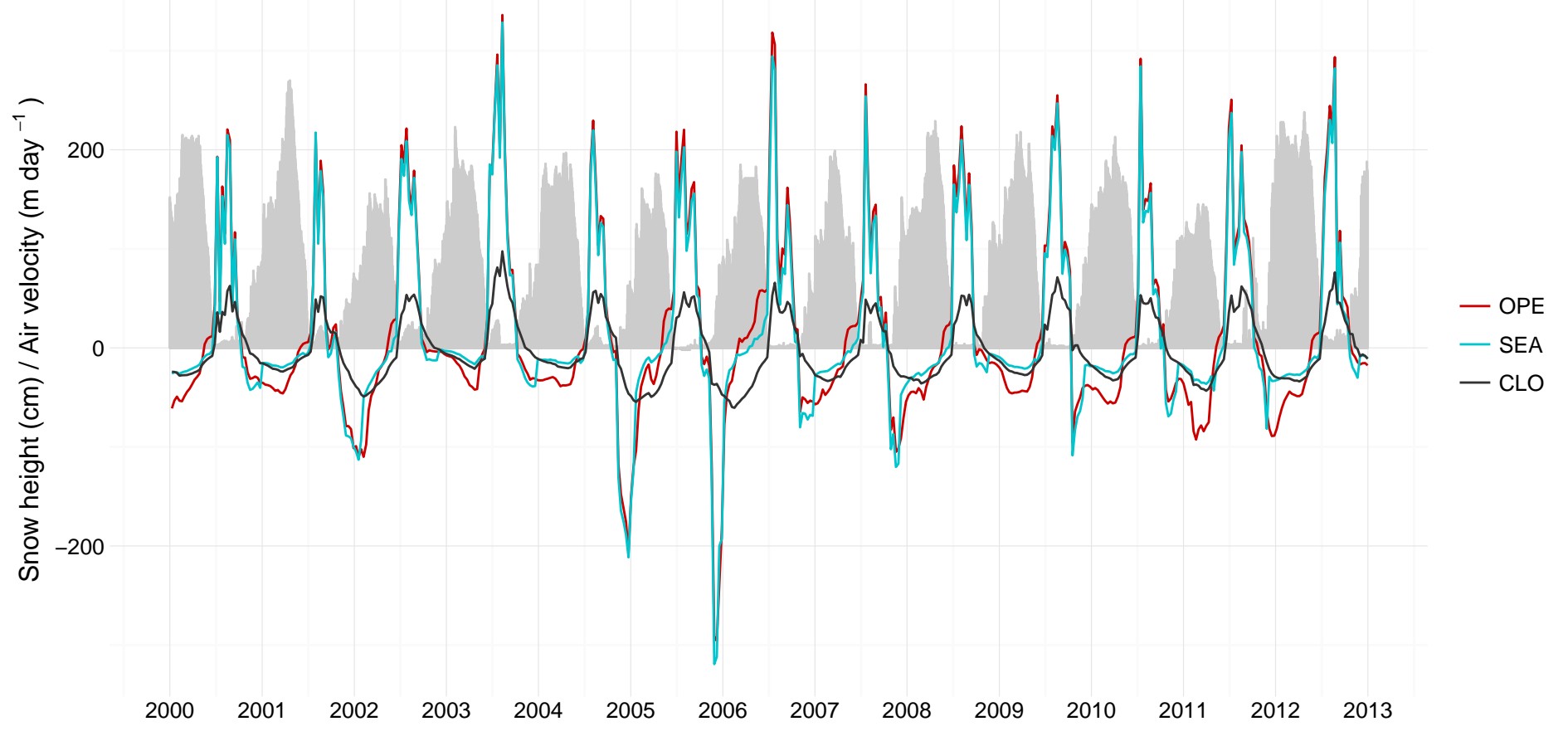

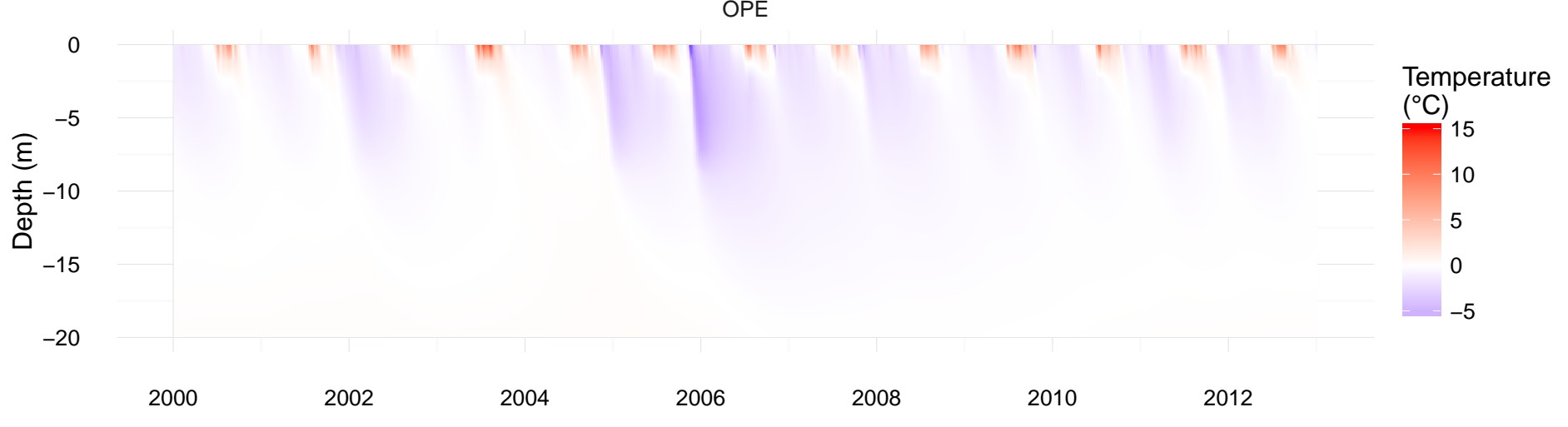

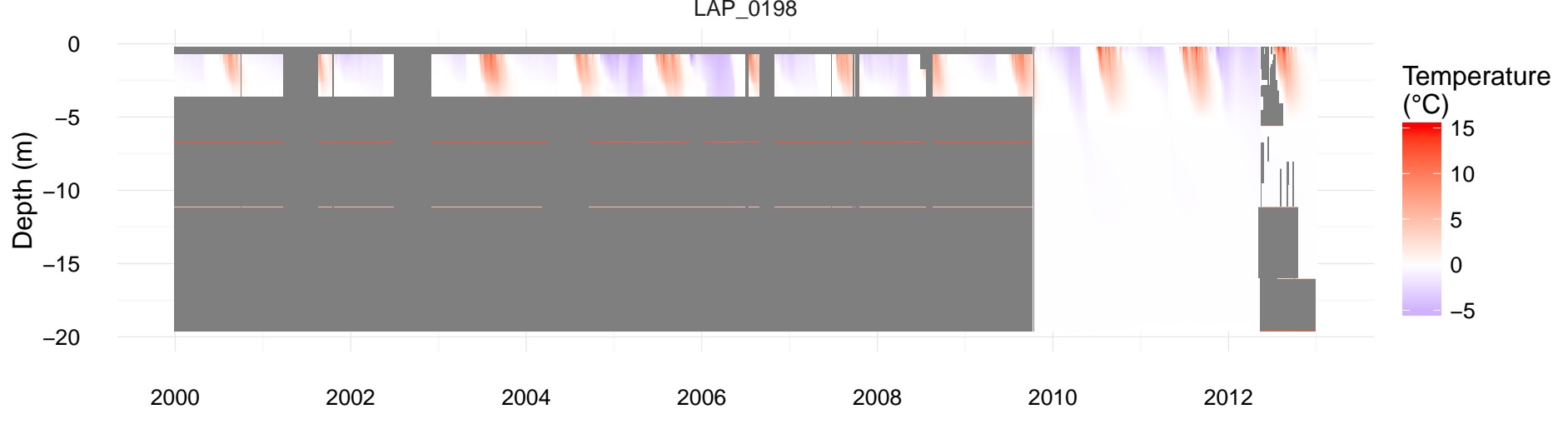

LAP_0198