# Peer review of "Numerical modelling of convective heat transport by air flow in permafrost talus slopes"

_The Cryosphere, 2016_

## Referee Comment (RC1) · Anonymous Referee #1 · 25 Nov 2016

In this study the authors successfully model convective flow in form of the so-called 'chimney-effect' in a permafrost affected talus slope. This is achieved in an idealised model setup using a commercially available model, GeoStudio. The authors simulate seasonably dependent flowpaths and cooling/warming effects at toe/head of slope. These results are in agreement with field observations from previous studies.

I think this is an interesting and valuable study that demonstrates, to my knowledge, a first numerical simulation of such effects in an alpine setting. It is a good first step towards explicitly describing this significant process in commonly used permafrost models. The study is clearly setup and manuscript well written with a nice flow of argumentation which made it enjoyable to follow the authors work. I think this study is a useful

contribution to field of permafrost modelling and only recommend the largely superficial comments below.

COMMENTS

1. Abstract l15: these numbers do not appear elsewhere in your text/results. Make sure consistent with main text.

2. p2 l3: you mean overcooled?

3. p5 l25: ...consists of gneiss and is at least 40 m deep, as observed in borehole cores...

4. p6 l2: "They were all able..." - Who were? –> "Such previous studies were able to show..."

5. p6 l3: "on a small scale" –> at fine scales.

6. p6 l16: remove "hereby".

7. p6 l19: "model runs numerically stable" –> model is numererically stable.

8. p7 l9: remove 'similar'.

9. p8 l24-28: Try to break this sentence up so more readable - quite a mouthful now.

10. p9 l2/Fig 4. Perhaqps mention/explain the bidirectional flow in June/July.

11. p9 l2: remove "hereby".

12. p9 l5: remove "hereby".

13. p10 l23: 'Identical'- Im being picky but they dont quite look identical - which could be true as you have some residual summer snow which would have an effect from 20cm depth. Of cause this summer snow could be measurment problems like the IMIS-grass effect.

14. p12 l1: "...comparison can only be qualitatively made..." –> ...comparison can only
be qualitatively made...

15. p12 l2: "Most notably is hereby the presence..." –> "Most notable is the presence..."

16. p12 l6-8: Section 5.3: refernce Figure 9 is missing.

17. p12 l18: Perhaps - The strength of this modelling approach lies in the fact that convective heat transfer is...

18. p12 l22: ..and hence there is no ice buildup...

19. p12 l22: This allows assesment of the influence...

20. p13 l9: we didnt see these values before. Similar to comment #1, be consistent with results in main text.

21. Acknowledgements: mention IMIS.

22. Table 1: remove hereby –> The snow layer is represented as an idealised...

23. Figure 6: Use solid/dashed lines, or similar, to distinguish nodes A/B.

---

## Referee Comment (RC2) · Lukas Arenson (Referee) · 2 Jan 2017

Dear Jonas and Christian,

First I'd like to offer my apologies for the delay in reviewing your manuscript. Unforeseeable circumstances did not allow me to review your contribution earlier. Once I finally started, I enjoyed reading your paper. I understand that there are significant limitations in modelling actual air movements and related ground temperatures caused by convective heat transfers in coarse talus slopes, the paper provides a good overview of why supercooling is observed in such materials. The paper is understandable and overall only requires some minor modifications. I have added comments and suggestions in the annotated version you find as a supplement file below. Also, please make sure

that you differentiate between natural (density driven) and forced (pressure difference) driven air convection. In such a slope, both types typically occur, however, your paper specifically focuses on gravity driven natural convection. Additional, more general comments:

- Page 3, Line 17: Make sure you add the limitation of modelling natural convection in 1D, i.e. that it isn't possible because you need at least 2-D to model the development of convection cells. It may help to describe why convective cells form and what determines their form.

- In Section 5.2. the Rayleigh number is mentioned for the first time without explanation. Please add it in the introduction and explain its significance when analysing convective heat transfer. You may want to link this to the point above.

- It is not clear why the air space has been included in the model, instead of applying an air pressure boundary conditions and air entry value (to allow air to move into the talus slope or not)

- In terms of material parameters chosen, it is understood that those may not have physical meanings per se, and shouldn't necessary be compared with what one would measure, however, in relation to each other some care must be taken. The following observations were made that may require an explanation by the authors in terms of how those values may affect the results:

o Thermal conductivity of air is much higher that actual thermal conductivity of air.

o Conductivity in air is in essence infinite, i.e. the model setup, where the value for the talus material was chosen to be the same, may affect the result.

- Page 8, line 27: It is not clear what is not homogeneous in this surface layer. If it is pure conductive heat transfer, the results should be homogeneous. If not, it is typically a sign for numerical instabilities.

- In the results and discussion section it would have been helpful to show the results

of the convective models in relation to the pure conduction model. This clearly shows where the air convection is the dominant process and to what extent.

- In the limitations a discussion on the snow cover would be helpful, i.e. the limits on how this is model, in particular when holes form that affect the overall air flow through the talus cone.

- Finally, in the conclusions, do not forget the hydrology, i.e. infiltration of water, in particular snow melt. There is additional advection from the infiltration of melt water that can affect the ground temperatures, but is also key when generating a frozen core.

Please also note the supplement to this comment:
http://www.the-cryosphere-discuss.net/tc-2016-227/tc-2016-227-RC2-supplement.pdf

**Supplement:**

[revised manuscript text omitted]

---

## Author Comment (AC1) · 10 Mar 2017

**Author responses to reviewer comments**

The responses are in *italic*.

**Comment from Anonymous Referee #1 ()**

In this study the authors successfully model convective flow in form of the so-called 'chimney-effect' in a permafrost affected talus slope. This is achieved in an idealised model setup using a commercially available model, GeoStudio. The authors simulate seasonably dependent flowpaths and cooling/warming effects at toe/head of slope. These results are in agreement with field observations from previous studies. I think this is an interesting and valuable study that demonstrates, to my knowledge, a first numerical simulation of such effects in an alpine setting. It is a good first step towards explicitly describing this significant process in commonly used permafrost models. The study is clearly setup and manuscript well written with a nice flow of argumentation which made it enjoyable to follow the authors work. I think this study is a useful contribution to field of permafrost modelling and only recommend the largely superficial comments below.

COMMENTS

1. Abstract l15: these numbers do not appear elsewhere in your text/results. Make sure consistent with main text.

> *That is right, and we apologise for the inconsistency. The numbers did not appear in the text and are an order of magnitude visible in Fig. 6 (for specific values cf response to comment 20 and 4.2 Temperature in the results section). We modified the sentence in the abstract as follows: "Modelling results show that convective heat transfer has the potential to develop a significant temperature difference between the lower and the upper part of the talus slope."*

2. p2 l3: you mean overcooled?

> *No, undercooled seems to be a common term to describe such thermal phenomena in the context of permafrost (for example: Stiegler et al. 2014).*

3. p5 l25: ...consists of gneiss and is at least 40 m deep, as observed in borehole cores...

> *Modified accordingly.*

4. p6 l2: "They were all able..." - Who were? –> "Such previous studies were able to show..."

> *Modified accordingly.*

5. p6 l3: "on a small scale" –> at fine scales.

> *Modified accordingly.*

6. p6 l16: remove "hereby".

> *Modified accordingly.*

7. p6 l19: "model runs numerically stable" –> model is numererically stable.

*Modified accordingly.*

8. p7 l9: remove 'similar'.

*Modified accordingly.*

9. p8 l24-28: Try to break this sentence up so more readable - quite a mouthful now.

*Modified accordingly.*

10. p9 l2/Fig 4. Perhaqps mention/explain the bidirectional flow in June/July.

*We do not really see the bi-directional flow on Fig. 4: within the talus a downward flow develops. However, the flow velocity actually is very different, in June the velocity is very low compared to July. In many of the years, the snow cover persists until June and therefore the gradient between the talus temperature and the forcing GST at the surface atmosphere boundary is lower than in the following (snow-free) months. We modified the sentence as follows: "The strongest circulation (snow free and therefore large temperature gradient at the atmosphere – talus boundary) is simulated in July-August whereas the winter circulation is less strong, but continues over a longer time period (7 months, in contrast to only 3-4 months in summer)."*

11. p9 l2: remove "hereby".

*Modified accordingly.*

12. p9 l5: remove "hereby".

*Modified accordingly.*

13. p10 l23: 'Identical'- I'm being picky but they don't quite look identical - which could be true as you have some residual summer snow which would have an effect from 20cmdepth. Of cause this summer snow could be measurement problems like the IMIS-grasseffect.

*We fully agree that these values are actually not identical. The slightly higher air velocity in the OPE experiment (Fig.8, red line) are the result of the more efficient winter ground cooling and thus due to the higher gradient more active summer circulation. The small differences during summer times are neither due to the "IMIS-grass effect" nor any kind of residual snow. The snow data used was linked to the conductivity as described in p.7 l.20-23 in the Discussion Paper: "For a snow height lower than 0.2 m the snow layer does not restrict the circulation. From a snow height of 0.2m to 0.8m air conductivity linearly decreases from $10^4$ to 0 m day-1 and thus makes the exchange between the air and the talus impossible for snow heights above 0.8 m (cf. Scherler et al. 2013)." We adapted the manuscript in section 5.1 Process analysis as following:  "In addition, velocities in summer for OPE and SEA show a similar behaviour as no snow cover is present. Due to the more efficient ground cooling in winter, the OPE experiment shows slightly higher air velocity values."*

14. p12 l1: "...comparison can only be qualitatively made..." –> ...comparison can only be qualitatively made...

*We are sorry but we do not really understand this comment. Both phrases are identical ?!*

15. p12 l2: "Most notably is hereby the presence..." –> "Most notable is the presence..."

*Modified accordingly.*

16. p12 l6-8: Section 5.3: refernce Figure 9 is missing.

*Modified accordingly.*

17. p12 l18: Perhaps - The strength of this modelling approach lies in the fact that convective heat transfer is...

*Modified accordingly.*

18. p12 l22: ..and hence there is no ice buildup...

*Modified accordingly.*

19. p12 l22: This allows assessment of the influence...

*Modified accordingly.*

20. p13 l9: we didnt see these values before. Similar to comment #1, be consistent with results in main text.

*We are sorry that we forgot to mention this in the results. Consequently, we integrated these values in the result section 4.2 Temperature in the revised manuscript adding the following sentence: "The mean temperatures in the lower part of the talus at node B over the 13 years modelling period decrease by 0.28 °C (CLO), 0.94°C (SEA) and 1.19°C (OPE), respectively, compared to the CON experiment with no convective cooling."*

21. Acknowledgements: mention IMIS.

*Modified accordingly.*

22. Table 1: remove hereby –> The snow layer is represented as an idealised...

*Modified accordingly.*

23. Figure 6: Use solid/dashed lines, or similar, to distinguish nodes A/B.

*We used dashed lines for the nodes B on Fig. 6 in the revised manuscript.*

**Comment from Lukas Arenson (Referee #2) ()**

Dear Jonas and Christian,

First I'd like to offer my apologies for the delay in reviewing your manuscript. Unforeseeable circumstances did not allow me to review your contribution earlier. Once I finally started, I enjoyed reading your paper. I understand that there are significant limitations in modelling actual air movements and related ground temperatures caused by convective heat transfers in coarse talus slopes, the paper provides a good overview of why supercooling is observed in such materials. The paper is understandable and overall only requires some minor modifications. I have added comments and suggestions in the annotated version you find as a supplement file below. Also, please make sure that you differentiate between natural (density driven) and forced (pressure difference) driven air convection. In such a slope, both types typically occur, however, your paper specifically focuses on gravity driven natural convection. Additional, more general comments:

- Page 3, Line 17: Make sure you add the limitation of modelling natural convection in 1D, i.e. that it isn't possible because you need at least 2-D to model the development of convection cells. It may help to describe why convective cells form and what determines their form.

> We are thankful for this comment and suggestion. We consequently included a new paragraph in section 2.1 (conceptual model) and rearranged the modelling examples accordingly. The new paragraph reads as:
> "In general, these convection cells form by natural convection, i.e. air movement as a function of density (temperature) differences, as opposed to forced convection due to e.g. the influence of surface (atmospheric) wind (e.g. Arenson and Sego 2007). Hereby, the air movement due to natural convection has to be large enough with respect to the bulk thermal conductivity of the material to yield a sustained convective cell. This relation is expressed by the Rayleigh number, which relates the air permeability, the thickness of the porous layer, the thermal conductivity of the material and the spatial temperature gradient within the porous layer, here the talus material. The higher the air permeability, the thickness of the talus and the temperature gradient with respect to the thermal conductivity, the higher the Rayleigh number and the stronger and spatially extensive the convective cell(s) (Arenson and Sego 2007). However, as some of the important parameter of the Rayleigh number may be temporally and spatially variable, the air circulation will change in space and time as well. Consequently, an explicit modelling of the air circulation in 2 dimensions is necessary to be able to simulate the development and seasonality of the occurrence of convection cells in talus slopes."

- In Section 5.2. the Rayleigh number is mentioned for the first time without explanation. Please add it in the introduction and explain its significance when analysing convective heat transfer. You may want to link this to the point above.

> See our answer to the comment above: we introduce now the Rayleigh number already in section 2.1 and explain its significance regarding the development of convection cells. A reference to this paragraph is made additionally in section 5.2.

- It is not clear why the air space has been included in the model, instead of applying an air pressure boundary conditions and air entry value (to allow air to move into the talus slope or not).

> *The air block has some advantages compared to a boundary pressure condition. First of all, the atmosphere surface boundary is not horizontal and therefore an altitude depending pressure gradient needs to be defined, leading to a more complex model formulation. Furthermore, the air entry (or exit) parameters would need a calibration process to obtain realistic and especially numerically stable results. Still, a follow-up study is planned and we will attempt to integrate a model set up with pressure boundary conditions to compare both. The text was modified accordingly, see below.*

- In terms of material parameters chosen, it is understood that those may not have physical meanings per se, and shouldn't necessary be compared with what one would measure, however, in relation to each other some care must be taken. The following observations were made that may require an explanation by the authors in terms of how those values may affect the results:

o Thermal conductivity of air is much higher that actual thermal conductivity of air.

> *We suggest that the much higher thermal conductivity of air in the model setup compensates to some extent the missing turbulent fluxes and the exchange with the atmosphere. The higher conductivity allows more pronounced exchange with the air block and therefore probably more realistic conditions. Furthermore a steep thermal conductivity gradient at the surface-atmosphere boundary is prone to numerical instabilities. The text was modified accordingly, see below.*

o Conductivity in air is in essence infinite, i.e. the model setup, where the value for the talus material was chosen to be the same, may affect the result.

> *We think that this affects the results in terms of absolute values but that the underlying process would not change in a significant way. The seasonally altering air circulation pattern within the talus would stay the same. The ground-cooling may will be more/less pronounced depending on how the boundary conditions at the surface atmosphere boundary will be set. But as many other parameters are subject to uncertainties and may affect the absolute values, we think the strength of this study is the representation of the underlying processes.*

> *To address the above three comments and answers regarding the simplified representation of air/atmosphere we added the following paragraph to the discussion section of the revised manuscript:*

> *"First of all, a pressure boundary condition could have been applied to link the talus slope to the atmosphere. Air flow at the boundary is not known and would have needed further parameterizations therefore this approach was dismissed. Secondly, we assume that the too high thermal conductivity of air used in this study compensates to some extent the missing turbulent fluxes which are prevented by the low air conductivity. This may affect the absolute values but allows numerically consistent simulations which represent the underlying process within the talus slope."*

- Page 8, line 27: It is not clear what is not homogeneous in this surface layer. If it is pure conductive heat transfer, the results should be homogeneous. If not, it is typically a sign for numerical instabilities.

> *We apologize for this imprecise formulation. The model geometry actually is slightly asymmetrical, which also leads to an asymmetric temperature distribution in the layers closer to the surface. As visible in Fig. 6 (CON A / CON B), the differences are marginal. We replaced the sentence with a comment on the asymmetry:" The small difference between node A and B in Fig. 6 is due to the slightly asymmetric model geometry."*

- In the results and discussion section it would have been helpful to show the results of the convective models in relation to the pure conduction model. This clearly shows where the air convection is the dominant process and to what extent.

> *We think that Fig.6 illustrates this influence in a good way. To illustrate the spatial variability we added a figure showing the spatial differences to the supplementary material and referred to it in the results section 4.2 Temperature: "The spatial variability of the temperature difference to the OPE experiment is shown in Fig. 10 in the supplementary material."*

- In the limitations a discussion on the snow cover would be helpful, i.e. the limits on how this is model, in particular when holes form that affect the overall air flow through the talus cone.

> *We added the following paragraph to the section "5.4 Models strength and weaknesses" to better illustrate the importance of the snow cover and the complex feedback effects that may result: "The representation of snow still is quite poor. The interactions between the talus slope circulation and snow layer are complex and not yet fully understood. Melt holes due to warm air exits are frequently observed at the top of a talus slope (Morard 2011) and probably have a significant impact on the air flow path and the air velocity. The pronounced ground cooling in the lower part may also influence the snow cover, especially at its base some refreezing of percolating melt water may take place."*

- Finally, in the conclusions, do not forget the hydrology, i.e. infiltration of water, in particular snow melt. There is additional advection from the infiltration of melt water that can affect the ground temperatures, but is also key when generating a frozen core.

> *We agree on the importance of water and mention it now in the revised manuscript not only in the model setup section and the discussion but also in the conclusions with the following sentence: "Furthermore the infiltration of melt water and precipitation as well as intra talus water flows, which can lead to advective heat transfer and thus have an influence on the ground thermal regime, are neglected."*

Please also note the supplement to this comment: http://www.the-cryosphere-discuss.net/tc-2016-227/tc-2016-227-RC2-supplement.pdf

> *Thank you very much for the precise revision, suggestions and corrections in the supplementary file. We modified the revised manuscript accordingly. Two points need a short explanation from our side:*

- p2/l3: supercooled instead of undercooled: *We kept the term "undercooled" to describe the thermal state of a talus slope; it seems to be a more common term to describe such thermal phenomena in the context of Alpine permafrost (for example: Stiegler et al. 2014).*
- p15/l11: Not clear how you use the difference between advection and convection in this case: *We used these two terms to differentiate between a dominantly vertical convection and dominantly horizontal (respectively parallel to the surface) convection, which is often also referred as advection. We added the term "horizontal" in the revised manuscript to clarify this issue.*

---

## Editor Decision (ED1)

[revised manuscript text omitted]

**Winter phase**
(«warm» air outflow involving an aspiration of cold air)

$T_{ao} < T_{ai}$

Snowpack

Zone with ground surface temperature >0°C

Zone of maximal winter overcooling

$T_{ao}$

$T_{ai}$

Substratum

Permafrost (?)

**Summer phase**
(gravity discharge of cold air)

$T_{ao} > T_{ai}$

Zone of minimum summer warming

Substratum

Talus slope

Permafrost (?)

[Figure]

[Figure]

[Figure]

[Figure]

[Figure]

[Figure]

[Figure]

[Figure]

[Figure]

LAP_0198

---

## Author Response (AR2)

**Author responses to editor comments**

We are very thankful for the comments and suggestions in the attached .pdf supplementary file and modified the manuscript accordingly. Below we provide responses to open question from the supplementary file. The responses are in *italic*.

p6, l5

"The porosity within the permafrost layer ranges from 30 to 60% and is, in some part of the talus slope, mostly sealed by ice (Scapozza 2013, Hilbich 2010)."

parts (where? at the base of the slope?)

> *Actually, in the Lapires talus slope ground ice is not only located at its base as one would expect. The distribution of ground ice and permafrost is quite complex and a detailed description is beyond the scope of this paper. Delaloye (2004) and Scapozza (2013) provide exhaustive information on the topic. We adapted the sentence:*
>
> *"The porosity within the permafrost layer ranges from 30 to 60% and some parts of the talus slope (including the base of the slope) are sealed by ice (cf. Scapozza 2013, Hilbich 2010)."*

P6, l27

meaning what? necessary for convergence?

> *The Courant criterion ensures that the distance that a fluid potentially travels within a time step does not exceed the distance between the different elements of the mesh. If the Courant criterion is not met, it is probable to run into numerical instabilities (see also GeoStudio, 2013a, c). To clarify this issue we adapted the sentence as follows:*
>
> *"The maximum time step is 0.5 days, decreasing to 0.1 days until the Courant criterion is met to minimize numerical dispersion and oscillation (GeoStudio 2013a, c)."*

P9, l14

Can you say anything about the temperature differences required for circulation to occur?

> *Even with (very) small temperature differences circulation occurs in the modelling results. However, a weak circulation has almost no thermal effects and the convection then only has a minor influence on the ground thermal regime. If this is also the case in reality or if there is a certain threshold value that triggers circulation remains an open question and should be addressed in a follow-up study. To be complete, we added the following sentence to the manuscript:*
>
> *"Even low temperature gradients cause a circulation in the modelling results. However, a weak circulation has almost no thermal effect and thus conduction dominates."*

P10, l14

"Temperature differences between the upper and lower part of the talus slope are up to 6°C during the warm and cold years 2003-2006, which is an effect of the air circulation alone, as no significant temperature differences were obtained in the conductive reference experiment (CON, grey and black line on Fig. 6)."

unclear... are you talking about 2003 AND 2006 or about all the years between 2003 and 2006 - and which is warm and which is cold? (I thought both 2003 and 2006 were warm)

> *We apologize for this imprecise formulation. We wanted to highlight the exceptionally warm summer in 2003 and the two cold early winters in 2004/05 and 2005/06 causing a high temperature gradient and thus an efficient air circulation. We changed the sentence to be clearer:*
>

[revised manuscript text omitted]

**Supplementary material**

[Figure]

**Figure 10: Simulated temperature difference (colours) in the talus slope between the open (OPE) and the conduction only (CON) experiment for day 300 (winter circulation).**